# MULTIPLE POSITIVE VIEWS IN SELF-SUPERVISED LEARNING

## ABSTRACT

Contrastive learning is a potent technique for self-supervised learning (SSL) that maintains invariance between two views. Advancements such as the "core view" (Tian et al., 2020a) or multi-cropping have harnessed insights from multiple views, culminating in the latest state-of-the-art performance. However, the complexities of multiview learning remain partially unexplored. In this paper, we introduce a "plug-and-play" multi-positive-views ($\geq 3$) learning approach seamlessly integrated with existing two-view SSL architectures. Theoretical and empirical analyses underscore the feasibility of enhancing traditional SSL models by incorporating multiple positive views. By mitigating the intrinsic biases towards sufficiency and minimality in the embeddings, our method achieves improvements in average accuracy (2% on CIFAR-10 and 26% on Tiny ImageNet) and significant speed-ups (3–4 times) across five datasets and eight architectures. Our research reveals and improves the double-edged nature of conventional assumptions tied to two-view suitability, thereby paving the way for future investigations in multiview SSL.

## 1 INTRODUCTION

Self-supervised learning (SSL) (Grill et al., 2020) is a powerful paradigm in representation learning, bringing about a host of functions and corresponding schemes that have immensely developed machine learning. The path we have traversed, from the inception of representation learning (Tian et al., 2020a), through self-distillation (Grill et al., 2020; Chen & He, 2021), correlation analysis (Zbontar et al., 2021), contrastive learning (Oord et al., 2018), to multiview invariance (Balestriero et al., 2023), illustrates a paradigm shift that has expanded our understanding of learning algorithms and has reshaped the landscape of machine learning research.

Previous research in the realm of SSL aimed to maximize mutual information between two views as a pretext task (Bachman et al., 2019; Oord et al., 2018). A data point can generate "positive views" and "negative views." As shown in Figure 1, positive views arise from identical transformations applied to the same image, whereas differences in images within the same batch constitute negative views. This process of "pushing" the views apart along both sample and feature dimensions serves to infuse diversity and variability, enabling models to acquire more perceptive and transferable representations (Shorten & Khoshgoftaar, 2019). However, when we refer to multimodal learning (Huang et al., 2021), it denotes leveraging various modalities (three or more).

Multi-view invariant means $2n - 2$ negative samples and only 1 positive pair, which naturally prompts the question, "Why not extend the idea of attracting $n$ positive samples in contrastive learning as well?" In fact, multiview ($\geq 3$) approaches are not a frontier concept in SSL. Recent advancements have been extensively influenced and benefited from the essence of multiple-positive learning. For instance, Bardes et al. (2022b); Tong et al. (2023); Caron et al. (2021) harnessed multi-cropping augmentation, blending multi-scale perspectives to obtain a more "global" insight. Drawing parallels with methodologies presented in (Tian et al., 2020a), these approaches implicitly incorporate positive views by simultaneously optimizing several pairs.

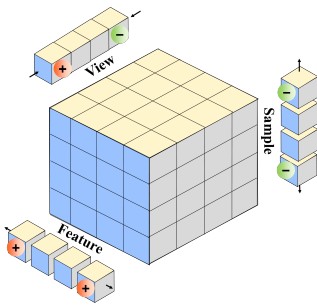

Figure 1: Three dimensions in self-supervised learning

The recurrent prominence of such methods in the latest state-of-the-art (SOTA) literature attests to their substantial potential in the SSL domain.

Recent advancements in this domain have motivated our investigation. We observe that existing methods present seemingly contradictory viewpoints, such as maximizing information representation in InfoMax (Linsker, 1988), as opposed to minimizing it in InfoMin (Tian et al., 2020b). Notably, several of these approaches are rooted in a myriad of complex assumptions, as highlighted by Balestriero et al. (2023). In response to identifying needs and gaps, we do not perceive these perspectives as entirely antagonistic. Instead, we critically examine these divergent viewpoints, highlighting the necessity to acknowledge and address the unique complexities introduced by multiview learning. Our key contributions to the SSL are as follows:

- **Theoretical Evidence Synthesis**: By challenging the traditional assumptions that all views share the same task information (Zhao et al., 2017), we establish the connection between multiview($\geq 3$) learning and bottleneck theory defined on downstream tasks.
- **Method Development**: We present a "plug-and-play" multiview ($\geq 3$) learning method, seamlessly integrating existing two-view SSL architectures and enhancing their ability to capture data diversity and richness.
- **Performance Improvement**: Our method improves accuracy (ranging from 2%-26%) and significant speed-ups (up to 18 times) across diverse datasets and architectures. It also excels in domain transfer learning scenarios.
- **Ablation Studies**: We examine how increasing the number of views affects optimization, considering factors such as data augmentation and batch size.

## 2 RELATED WORK

**Objective of SSL** Unified SSL paradigms emerged relatively late (Shwartz-Ziv et al., 2022; Balestriero et al., 2023). Previously, SSL drew inspiration from contrastive, metric, and multiview learning. Therefore, optimization objectives can be obtained from background information in the relevant domain. For example, InfoMax, a reconstruction-based method (Linsker, 1988), emphasizes maximizing information retention (Zhang et al., 2016; Srivastava et al., 2015; Gidaris et al., 2018). Conversely, model compression such as InfoMin (Tian et al., 2020b) considers the noise implications on the accuracy of downstream tasks. Methods such as pretext-invariance (Misra & Maaten, 2020), model distillation (Tarvainen & Valpola, 2017), and information bottleneck (Federici et al., 2020) achieve this.

Training samples can be categorized into inter-class label or label-free positive-negative pairs. Metric learning primarily focuses on maximizing inter-class variance with techniques such as triplet loss (Schroff et al., 2015; Sohn, 2016). Contrastive learning, bearing similarities with metric learning, takes positive-negative pairs as input, with prominent models being SimCLR (Chen et al., 2020); noise contrastive estimation (NCE), such as DIM (Hjelm et al., 2019); and InfoNCE variants, such as contrastive predictive coding (CPC) (Oord et al., 2018) and MoCo (He et al., 2020). These paradigms aim to distinguish diverse data samples through embeddings. Notably, CPC, being emblematic of contrastive objectives, is a lower bound on mutual information, which may indicate a significant shift toward the view-invariance method termed by (Balestriero et al., 2023).

**Multiple Positive Views** While dedicated SSL methods with multiple positive views are scarce, insights can be gained from multiview learning studies that provide both theoretical foundations (Geng et al., 2020) and empirical evidence (Tsai et al., 2020). The multiview criteria include: i. Consensus Principle and ii. Complementarity Principle (Xu et al., 2013), which in the SSL domain translates into minimal sufficiency and mutual redundancy (Shwartz-Ziv & LeCun, 2023; Tian et al., 2020b), respectively. Under these premises, DMVIB and CMC Wang et al. (2019); Tian et al. (2020a) have validated the feasibility of similar work in both supervised and unsupervised multiview settings. Specifically, CMC proposes two training paradigms based on a "core view" defined as $\sum \text{loss}(z_0, z_{\text{mean}})$ and "global graph" characterized by $\frac{2}{n(n-1)} \sum \sum \text{loss}(z_i, z_j)$. Following these paradigms, SwAV (Caron et al., 2020), DINO (Caron et al., 2021), VICRegL (Bardes et al., 2022b), and EMP-SSL (Tong et al., 2023) were proposed, emphasizing the extraction of different insights from various views using multi-crop techniques to introduce local features as an additional dimension for enrichment. Moreover, a strategy called multiview clustering exists. This approach does not

explicitly rely on negative samples; instead, it harnesses complementary information from multiple views to derive common representations of multiple positive samples, as detailed in (Caron et al., 2020; Ermolov et al., 2021).

However, the view independence assumption might be overly stringent for both the multimodal and multiview domains. Research by Federici et al. (2020) suggests that contrastive learning cannot scale beyond two views, as previous studies did not consider the complexity of information interactions stemming from generalizing over two variables (Watanabe, 1960; Te Sun, 1980). Therefore, multiview learning (Isabelle et al., 2002) relaxes the independence assumption, adopting a co-training strategy with only weak dependencies.

## 3 METHOD

Grounded in Information Bottleneck theory, we assume that previous models aimed to maximize mutual information between augmented view pairs. Upon scrutinizing this goal, we identify several challenging conditions required for optimality. By relaxing these strict assumptions, we decoupled the objective from view-specific scenarios, enabling it application in multiview contexts.

### 3.1 VIEW-INVARIANT THEORETICAL OBJECTIVES FOR SSL

SSL creates high-quality embeddings or latent representations for downstream tasks such as classification or detection, regardless of the number of views (Chen et al., 2020; He et al., 2020; Chen & He, 2021). This process involves preserving task-specific information while filtering unnecessary data (Oord et al., 2018). This concept is expressed in the information bottleneck (IB) principle (Tishby et al., 1999), which is summarized in the following equation:

$$\min_{p(z|v)} I(V;Z) - \beta I(Z;Y) \tag{1}$$

where $V$ is the input, $Y$ is the output or labels, $Z$ denotes the representations, and $\beta$ is a positive trade-off parameter that is not inherently bounded. However, the IB principle (Tishby et al., 1999), originally devised for representation learning, introduces the entanglement between minimality $I(V;Z|Y)$ and sufficiency $I(V;Y|Z)$ in its formulation. However, even after optimizing for minimal sufficiency via the IB principle, as shown in Figure 2, the embeddings are prone to sufficiency bias that stems from incomplete coverage of view-specific and task-related information.

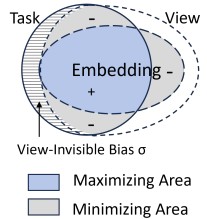

Figure 2: Optimization of minimal sufficiency

Therefore, assumed that the embedding $Z$ serves as the representation of the image $V$ (see appendix 8.4.1), we reworked the objective function on the left-hand side of Eq. (2). Our objective can be considered a lower bound of the IB objective. However, the two terms in our objective are independent, thus separating detrimental and beneficial information without the need for a trade-off parameter,

$$\min_{p(z|v)} I(V;Z|Y) - \beta I(Z;Y) + \sigma \overset{21}{=} \min_{p(z|v)} I(V;Z) - (\beta+1)I(Z;Y) + \sigma \tag{2}$$

$$\overset{22}{\propto} \min_{p(z|v)} \underbrace{I(V;Z|Y)}_{\text{Minimality bias}} + \underbrace{\beta I(V;Y|Z) + \sigma}_{\text{Sufficiency bias}} \tag{3}$$

where the terms $I(Z;Y)$ and $I(V;Y|Z)$, both integral parts of $Z$, are in a mutually exclusive state owing to the invariant nature of $I(V;Y)$ in optimization. Moreover, we define optimizable error terms as view-specific bias. Correspondingly, we introduce view-invisible bias $\sigma$ as follows:

1. **View-Specific Bias**: This bias originates from improper optimization within individual views, leading to deviations from minimal sufficiency. Thus, each view can independently mitigate this type of bias through its own optimization process.

2. **View-Invisible Bias** $\sigma$: Illustrated as the horizontal strip area in Figure 2, this term reflects the inherent error in a specific view that is unobservable and therefore cannot be optimized within that view alone. However, this type of bias can be alleviated by introducing additional prior knowledge or views, providing a more comprehensive representation.

## 3.2 PRETEXT TASK DESIGN: FROM TWO-VIEW TO MULTIVIEW LEARNING

### 3.2.1 REVISITING CONTRASTIVE LEARNING

The introduction of a corrected objective function in the previous subsection reveals the limitations within traditional contrastive learning methodologies (Oord et al., 2018; Shwartz-Ziv & LeCun, 2023). The classical methods often relied on idealized and sometimes conflicting assumptions, leading to biases and inefficiencies. These shortcomings were essentially overlooked because of the lack of a corrective objective function.

| **Hypothesis for single view** | | | **Mutually extended** | |
|---|---|---|---|---|
| Minimality | $I(v_i; z_i \mid y) = 0$ | $\rightarrow$ | $I(v_i; z_j \mid y) = 0$ | |
| Sufficiency | $I(v_i; y \mid z_i) = 0$ | $\rightarrow$ | $I(v_i; y \mid z_j) = 0$ | |
| Redundancy | $I(z_i; y \mid v_i) = 0$ | $\rightarrow$ | $I(z_i; y \mid v_j) = 0$ | |

Figure 3: Hypothesis in previous methods (Left); view of optimization (Right)

As shown in Figure 3, prior approaches focused on minimizing $I(z_1, z_2)$ assume that the two views share solely task-relevant information and are independent of redundancy information, while mutual counterparts broaden these principles to include mutual redundancy (Federici et al., 2020), minimality, and sufficiency (Tian et al., 2020b). Therefore, a link between $I(z_1; z_2)$ and the theoretical objective function can be established **only under an exceedingly strict set of conditions above**, serving as limiting constraints that may hinder stability and consistency.

$$\text{InfoNCE} \leq I(z_1; z_2) \tag{4}$$

$$\overset{23}{=} I(z_1; y) - I(z_1; y \mid z_2) + I(z_1; z_2 \mid y) \tag{5}$$

$$\propto -I(v_1; y \mid z_1) - I(z_1; y \mid z_2) + I(z_1; z_2 \mid y) \tag{6}$$

The issues of rigor can be further depicted in Figure 3, given that ideal assumptions are nearly impossible to fulfill owing to random fluctuations of the positive sample and augmentation. Consequently, the effective area, called the "safety zone" or "sweet spot" (Tian et al., 2020b), becomes smaller, leading to more invisible sufficiency biases and even view-specific biases. This discrepancy grows with increasing views, demonstrating the limitations of extending more views.

### 3.2.2 RECONSTRUCTION OF THE MULTIVIEW PRETEXT TASK

To address the underlying biases that can lead to poor convergence of embeddings, we initially set aside the hypotheses discussed in the previous section. Then we maximize mutual information (MI) between multiple views and a mean embedding denoted as $z_{\text{mean}}$. This principle is inspired by the concept of "embedding distillation," as highlighted in works such as mean shift (Koohpayegani et al., 2021) and MixMatch (Berthelot et al., 2019). Subsequently, we extend the sum of MI to achieve a "circular symmetric" distribution, especially when considering an increased number of views $\{v_1, \ldots, v_n\}$. Similar to contrastive learning in Eq.(6), the final formulation is as follows:

$$\boldsymbol{\Omega'} = \sum I(z_i; z_{\text{mean}}) \leftarrow \textit{pretext task} \tag{7}$$

$$\propto \sum I(v_i; y \mid z_i) - \sum I(z_i; y \mid z_{\text{mean}}) + \sum I(z_i, z_{\text{mean}} \mid y) \tag{8}$$

$$\propto \sum \underbrace{I(v_{\text{joint}}; y \mid z_i)}_{\text{Sufficiency Bias}} + \sum \beta \underbrace{I(v_{\text{joint}}; z_i \mid y)}_{\text{Minimality Bias}} - \sum \underbrace{I(z_{\text{joint}}; y \mid z_{\text{mean}})}_{\text{Decaying error}} \tag{9}$$

$$\text{s.t.} \quad I(z_1; \ldots; z_n) \leq H(z_{\text{mean}}) \leq H(z_1, \ldots, z_n) \tag{10}$$

The achievement of optimal embedding involves maximizing the minimum sufficient **joint embedding** across views, denoted as $I(z_1, z_2, \ldots, z_n; y)$, while concurrently minimizing joint redundancy. We elaborate on how this transition occurs from Eq. (8) to Eq. (9) in the subsequent points.

**i). Reducing Sufficiency Bias via Pseudo/Mean Embedding**

During optimization, the sufficiency bias can be decomposed into two components: view-specific sufficiency bias $I(v_i; y|z_i)$ and invisible sufficiency bias $H(y) - I(y; v)$. In our approach, view-specific sufficiency bias is already well-accounted for through Eq. (8). To address view-invisible bias, we propose a computationally efficient alternative to joint embedding by calculating the average embedding. This idea draws parallels with 1x1 convolution (Lin et al., 2013). The averaging operation serves to reinforce the strong, task-relevant features $Y$, as these features are unlikely to nullify each other. Conversely, irrelevant or view-specific features $H(V \mid Y)$ are more likely to be attenuated during this process. Therefore, we assert the constraint in Eq. (10) as follows:

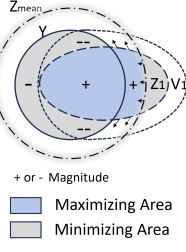

Figure 4: Pseudo embedding

$$I(z_1; \ldots; z_n) \leq H(z_{\text{mean}}) \leq H(z_1 \ldots z_n)$$

This allows us to deduce that $I(z_i; y \mid z_{\text{mean}})$ is a "subset" of $I(v_{\text{joint}}; y \mid z_i)$, reweighting the sufficiency term while preserving the direction.

**ii). Mitigating Minimality Bias via Forgetting Mechanism**

The minimality term is not explicitly optimized in the InfoNCE loss function and even experiences counter-optimization owing to the term $I(z_i, z_j \mid y)$. The introduction of multiple views and mean embedding implicitly resolves this issue, which can be understood in two iterative steps:

1. *Stabilizing the Direction of the Error Term*: As illustrated in Figure 4, with the increase in the number of views and iterations, the counter-optimizing gradients gradually decay, and the term $I(z_i, z_{\text{mean}} \mid y)$ naturally converges around the mean embedding in our approach. Consequently, the issue of unstable growth towards the uncompressed joint-view seen in contrastive learning during the iteration with changing view pairs is mitigated.

2. *Influence of Catastrophic Forgetting*: This stability gives rise to an interesting phenomenon in neural networks known as "catastrophic forgetting" (McCloskey & Cohen, 1989; Kirkpatrick et al., 2017). The embedding comprises three distinct types of information. Specifically, when a neural network optimizes only two types of terms—task-relevant visible and invisible sufficiency terms, it may inadvertently suppress or degrade its ability to represent task-irrelevant redundancy. This phenomenon is equivalent to implicitly introducing the slow-optimizing term $\beta I(v_{\text{joint}}; z_i \mid y)$, forming a closed-loop process[1].

### 3.3 GENERALIZATION TO ANY LOSS FUNCTION

Our method can be extended to most mutual information-based loss functions $\ell(z_i, z_j)$, and we derive the enhanced loss function integrated multiview strategy $L_m$ as follows :

$$L_m(\mathbf{z}_1, \mathbf{z}_2, \ldots, \mathbf{z}_n) \triangleq \sum_{i=1}^{n} I(\mathbf{z}_{\text{mean}}, \mathbf{z}_i) \geq \sum_{i=1}^{n} \ell(\mathbf{z}_{\text{mean}}, \mathbf{z}_i) \qquad (11)$$

where $\mathbf{z}_{\text{mean}} = \frac{1}{n} \sum_{i=1}^{n} \mathbf{z}_i$. For instance, if we consider the loss function of SimCLR, $l(z_i, z_j)$, it is often regarded as lower bounds of the mutual information (MI) between two variables (Chen et al., 2020; Oord et al., 2018), which establishes a connection with and benefits from our strategy.

Table 1: Comparison of multiview strategies in recent research (Config: Appendix 9.5.1)

| Strategy | Accuracy | GPU | Time | Integration of Multiview[2] | | Recent Research |
|---|---|---|---|---|---|---|
| Two Views | 49.5±0.2 | 0.47 | 28.2 | $\text{loss}(z_i, z_j)$ | | Contrastive Learning |
| FastSiam | 67.6±0.2 | 0.62 | 38.2 | $\frac{1}{n} \sum_i \text{loss}(z_i, \sum_{j \neq i} \frac{z_j}{n-1})$ | | (Pototzky et al., 2022) |
| Full Graph | 62.4±0.4 | 0.65 | 39.2 | $\sum_i \sum_{j \neq i} \text{loss}(z_i, z_j)$ | | (Tian et al., 2020a) |
| Core View | 44.6±0.6 | 0.66 | 38.0 | $\frac{1}{n-1} \sum_i \text{loss}(z_1, z_i)$ | | (Tian et al., 2020a) |
| Mean Shift | 54.9±0.7 | 0.61 | 33.4 | $\text{loss}(z_1, z_{\text{mean}})$ | | (Reiss & Hoshen, 2023) |
| SwAV | 64.8±0.4 | 0.63 | 35.4 | $\sum_{i \in \{1,2\}} \sum_{j \neq i} \text{loss}(z_i, z_j)$ | | (Caron et al., 2020) |
| Mean vs. N | 68.2±0.2 | 0.66 | 34.7 | $\frac{1}{n} \sum_i \text{loss}(z_i, z_{\text{mean}})$ | | Our Method |

[1]The contraction of $v_{\text{joint}}$ iteratively results in the inward migration of $z_{\text{mean}}$ boundary untill converge.

[2]Table note: gray circle ($z_i$), blue circle ($z_{\text{mean}}$), solid line (loss).

By mitigating both sufficiency and minimality biases, our approach outperforms all existing multiview methods. From the standpoint of regularization[3], our method effectively increases the Euclidean distance along feature and batch axes while concurrently reducing it along view axes.

## 4 EXPERIMENT

We extensively assessed the relative improvement and efficiency of our theory across five datasets and various model architectures. We ensured that all models and modifications shared the same configurations and further performed domain transfer to validate the transferability.

### 4.1 UNIFIED EXPERIMENT SETTINGS

We selected a lighter architecture, utilizing the backbone of ResNet-18 (He et al., 2016) and a batch size of 512 for consistency and efficiency across multiview configurations. This choice enables broader baseline comparisons within computational limits and enriches the significance for future work. To validate our theoretical assumptions, we adopted unselected data augmentations that slightly deviate from a "Sweet Spot" (Tian et al., 2020b). This setup shows that our method maintains a performance that is consistent with SOTA benchmarks, under less-tailored conditions.

### 4.2 BENCHMARK COMPARISON: BETWEEN TWO AND FOUR VIEWS

**Accuracy and Stability**: We compared the performance of models trained using both the original method and proposed technique, employing KNN accuracy as the evaluation metric[4]. As Table 2 presents, evident increases in accuracy were observed in most results. Moreover, our approach could prevent model collapse for specific models such as VICReg (Bardes et al., 2022a) under given conditions. Such improvement is attributed to the consideration of sufficiency and minimality bias in our method rather than relying on mere intuition. Notably, we found that the magnitude of improvement depends on the model to some extent. For example, NNCLR (Dwibedi et al., 2021) on CIFAR-10 and Tiny ImageNet benefit greatly from our method, resulting in relative improvements of 1.42% and 60.25%, respectively. Similarly, methods such as DCL (Yeh et al., 2022) that perform poorly on a smaller dataset are highly probable to fail on larger datasets as well. Theoretically, methods that strive to maximize MI may be more successful;

Table 2: **Accuracy comparison on CIFAR-10 and Tiny ImageNet** KNN accuracy was evaluated for eight different SSL architectures, employing two views (baseline) and four views (our method). We deliberately selected CIFAR-10 (Krizhevsky et al., 2009) and TinyImageNet (Le & Yang, 2015) considering the differences in resolution diversity and distinct data sources, serving to accentuate the stark contrast in class variety.

| Methods | Baseline (2 views) | Multiview (4 views) | Baseline (2 views) | Multiview (4 views) |
|---|---|---|---|---|
| BYOL Grill et al. (2020) | 91.31% | 92.39% | 36.31% | 43.76% |
| MoCo He et al. (2020) | 89.85% | 90.84% | 41.23% | 42.48% |
| SimCLR Chen et al. (2020) | 88.89% | 91.33% | 37.83% | 43.48% |
| SimSiam Chen & He (2021) | 90.13% | 90.50% | 27.96% | 42.24% |
| VICReg Bardes et al. (2022a) | 71.06% | 72.24% | Collapse | 33.31% |
| DCL Yeh et al. (2022) | 87.49% | 87.31%↓ | 40.14% | 38.44%↓ |
| TiCo Zhu et al. (2022) | 79.47% | 82.89% | 41.94% | 42.98% |
| NNCLR Dwibedi et al. (2021) | 89.07% | 90.34% | 24.38% | 39.07% |

Specifically, SwAV (Caron et al., 2020) reports an accuracy of 85.07% and 35.39%, while DINO (Caron et al., 2021) achieves 91.74% and 35.77%. In contrast, the improved baselines not only matches but in some cases exceeds the state-of-the-art (SOTA) architectural designs.

---

[3]When treating regularization as a linear concept, KL divergence and distance functions cannot be considered equivalent. However, they share the commonality of quantifying discrepancies.

[4]We adhere to the procedure outlined by Caron et al. (2020); Wu et al. (2018), akin to linear evaluation

**Efficiency and Performance**: As presented in Table 3, we rigorously evaluated the efficacy of the proposed method with focus on data exposure and convergence time. The results underscore a considerable advantage of our methodology, which is achieved by regulating the total number of views to maintain consistent data exposure. Moreover, our findings demonstrate a marked improvement in convergence efficiency. Our method achieves 90% and 95% of its optimal precision in fewer epochs, leading to speed increases by factors of 4.0 and 3.1, respectively. However, the reported improvements exhibit large standard deviations in ratio, indicating a notable variation across different datasets. Additionally, these improvements come with increased computational demands. The utilization of four views resulted in a 60.5% increase in GPU usage and a 92.6% increase in the average total runtime, characterized by 112.5% augmentation in the backward computation time.

Table 3: This table delineates the average relative improvement in KNN accuracy with same view data visibility (400 epochs for 4 views, 800 epoches for 2 views) and the efficiency in terms of speed-up ratio to reach a predetermined accuracy threshold during an 800 epoch training cycle, observed across **five datasets (STL-10, CIFAR-10, CIFAR-100, Tiny ImageNet, and ImageNette)**. Moreover, it documents the computational and GPU utilization costs incurred in the ImageNette.

| | Efficiency | | | Performance | | | |
|---|---|---|---|---|---|---|---|
| Method | 800 views | 90% | 95% | GPU | Backward | Forward | Total Time |
| BYOL | 10.8% (10.5) | 5.8 (3.8) | 4.7 (2.7) | 73.1% | 97.2 % | 1.0% | 86.5% |
| MoCo | 1.8% (3.8) | 1.9 (1.0) | 1.5 (0.7) | 65.6% | 103.2% | 2.7% | 87.6% |
| SimCLR | 7.1% (6.3) | 3.1 (1.6) | 2.1 (1.0) | 70.4% | 117.8% | 8.4% | 86.9% |
| SimSiam | 31.1% (29.3) | 9.1 (7.0) | 7.3 (6.5) | 39.6% | 84.2% | 28.4% | 84.6% |
| VICReg | 32.3% (43.0) | 6.4 (7.1) | 4.6 (4.9) | 42.6% | 168.3% | 3.0% | 101.8% |
| DCL | -2.9% (2.0) | 1.4 (0.8) | 1.1 (0.9) | 63.4% | 136.5% | 18.8% | 89.5% |
| TiCo | 2.1% (2.1) | 2.1 (1.1) | 1.6 (0.7) | 72.3% | 127.3% | 15.2% | 118.5% |
| NNCLR | 14.2% (27.6) | 2.2 (0.7) | 1.8 (0.8) | 57.3% | 65.3% | 14.2% | 85.1% |
| Average | 12.1% (15.6) | 4.0 (2.9) | 3.1 (2.3) | 60.5% | 112.5% | 11.5% | 92.6% |

## 4.3 VIEW REGULARIZATION

Figure 5 shows three types of analytical learning curves for DCL, and serves as a microcosm across all architectures, each consistently exhibiting a similar pattern. The plot underscores that our method functions as a regularization across a range of datasets and architectures by tuning and balancing three directions: view, batch, and feature (refer to 3.3). Furthermore, our method does not interfere with the model's intrinsic optimization landscape. However, similar to mini-batch techniques, our method markedly smoothed the loss function curve, reducing oscillations and preventing potential model collapse by balancing the optimization along three dimensions.

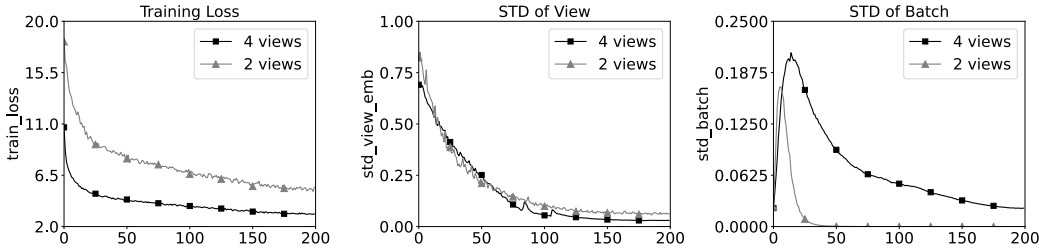

Figure 5: Curve of training loss, standard deviation over view and batch dimension

## 4.4 DOMAIN TRANSFER

Instead of task transfer such as segmentation or detection (Bardes et al., 2022a), we evaluated the improvement that can be realized across different domains (Patacchiola & Storkey, 2020). As depicted in Figure 6, our approach consistently outperformed the others in most of the results. However, these improvements do not seem consistent across disparate datasets or models. Moreover, an intriguing

cancellation symmetry was observed, where the transfer from ImageNette to Cifar10 was easily achieved, whereas the reverse seemed to induce inhibition to some extent. This symmetry holds only for improvements, not for original accuracies (for more information, please refer to Appendix 11.1). This observation implies a potential underlying complexity in multiview domain transfer, which may extend the scope of discussions beyond the current study.

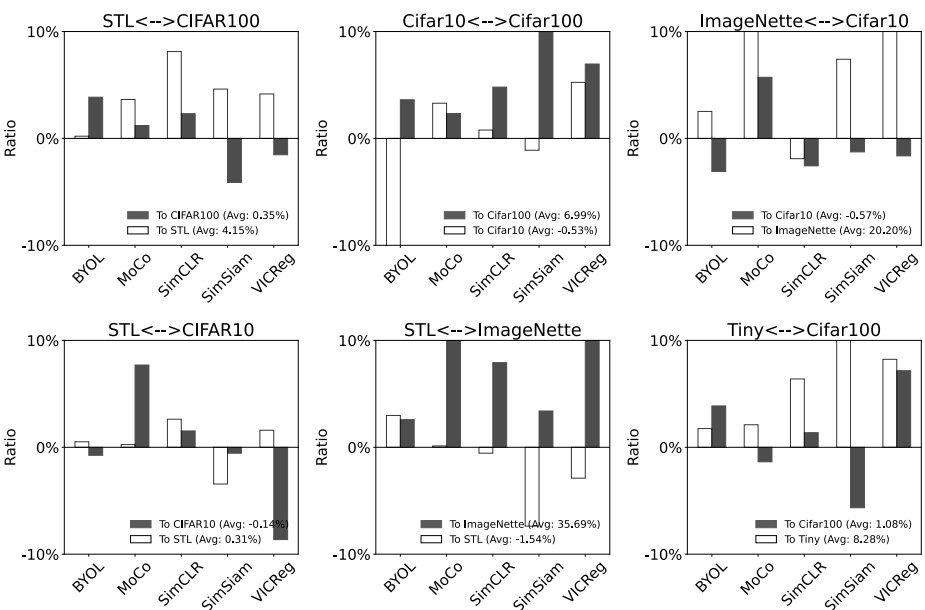

Figure 6: **Examination of models' domain transferability over five diverse datasets.** Training was conducted on the first indicated dataset, followed by linear evaluations on the second dataset (indicated with a white bar) and vice versa (indicated with a black bar). Any gains exceeding $\pm 10\%$ are capped at $10\%$. For further details on the implementation, please refer to Appendix 9.5.3.

# 5 ABLATION ANALYSIS

## 5.1 MARGINAL BENEFIT OF VIEW NUMBER

To determine the optimal number of views, we conducted tests ranging from two to ten views, utilizing identical settings (for further information, please refer to section 10.1). We compared two representative methods, MoCo and VICReg, as shown in Figure 7 and Figure 8. As the number of views increased, we observed a reduction in both variance and instability, leading to enhanced accuracy. This trend adheres to a logarithmic curve, signifying an optimal performance–accuracy trade-off between 4 and 6 views. The results align with the findings of Tian et al. (2020a) and contradict those of Pototzky et al. (2022), primarily because they do not account for the architecture-dependent nature of batch size optimization patterns.

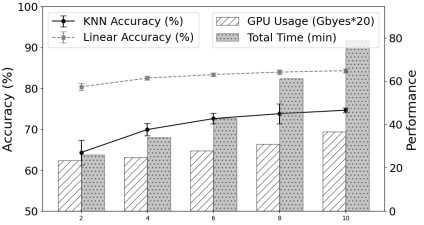

Figure 7: Performance of MoCo across views

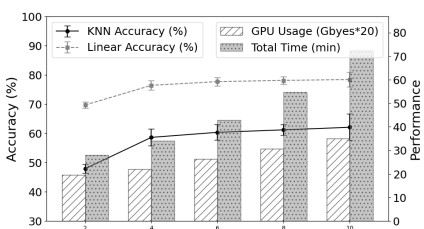

Figure 8: Performance of VICReg across views

## 5.2 BATCH SIZE

Contrary to previous research (Pototzky et al., 2022), it is evident that the use of multiple views does not change the optimization landscape of the original methods. In Table 4, our results reveal an extremely high correlation of batch series between the two- and four-view strategies for given epochs across batch sizes ranging from 32 to 2048. Using SimSiam as the underlying architecture, we found that our multiview strategy maintains high consistency with the original method, smoothens the optimization hyperplane, and improves the overall accuracy. Because mean embeddings stabilize oscillations in the two-view objective, while minimizing sufficiency and redundancy biases.

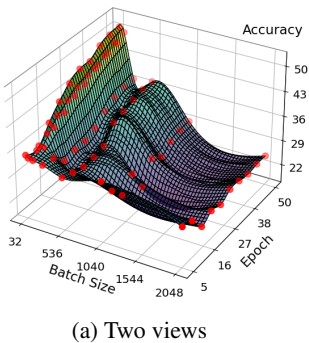

(a) Two views

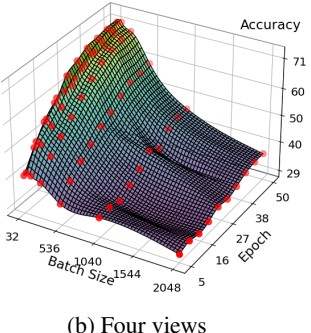

(b) Four views

| Correlations | | | |
|---|---|---|---|
| Epoch 0-25 | | Epoch 25-50 | |
| 5 | 0.83 | 30 | 0.86 |
| 10 | 0.99 | 35 | 0.90 |
| 15 | 0.93 | 40 | 0.91 |
| 20 | 0.88 | 45 | 0.91 |
| 25 | 0.90 | 50 | 0.91 |
| Short Avg:0.91 | | Long Avg:0.90 | |
| Total Average:0.90 | | | |

Table 4: Correlations along batch axis between (a) and (b)

## 5.3 AUGMENTATION ENSEMBLE

**Views Generated by Same Augmentation:** We first measured the accuracy of SimSiam in two-view scenarios ordered by the intensity of augmentation. The results followed the InfoMin principle (Tian et al., 2020b), demonstrating an inverted U-shaped curve. As the intensity of the view increased, our technique consistently improved in terms of accuracy. However, the original methods demonstrated unpredictable fluctuations, especially when employing weakly selected augmentations that align more closely with real-world scenarios.

**Views Generated by Different/Ensemble Augmentation:** In Table 5, each augmentation is divided into several sub-augmentations, e.g., cropping or color jitter only. All sub-augmentations lead to model collapse. Nevertheless, our method achieved better accuracy compared with the original method; however, it was lower than using four views with the same augmentations.

Table 5: Accuracy with different levels of augmentation strength

| Augmnemtation | View 1 | View 2 | View 3 | View 4 | View 5 |
|---|---|---|---|---|---|
| | | Weak —————— Strong | | | |
| 2 Views | Collapse | 41.7 | 40.9 | 36.9 | 49.3 |
| 4 Views | Collapse | 49.2 | 50.2 | 51.2 | 78.4 |
| **Ensemble** | | **30.5** | **38.2** | **41.6** | **49.7** |
| Transformation | ColorJitter | View 1 +Cropping | View 2 +Rotation | View 3 +RandomFlip | View 4 +Random Grayscale |

## 6 CONCLUSION

The conventional double-edged assumptions have both propelled success and constrained further advancements in SSL. In this paper, we propose a multiview-based method, grounded in the IB theory. Our method maximizes joint MI across multiple views by leveraging circular symmetry loss and pseudo-embedding techniques. Extensive validations across multiple models and datasets confirm its robustness and effectiveness, while also indicating a correlation between improved performance and an increased number of views. This study reveals the importance of multiview in reducing sufficiency and redundancy biases, setting the stage for future research.

# 7 REPRODUCIBILITY STATEMENT

To empirically validate the effectiveness of our proposed multiview SSL approach, we offer a quick pathway for the reproduction of the experiments outlined in this manuscript.

**1. Plug-and-Play Integration**    Our proposed methodology offers seamless integration into pre-existing architectures that are tailored for view-invariant tasks. We make only minimal adjustments to the existing codebase, thereby preserving the original architecture's integrity. Empirical evidence from our experiments demonstrates robust performance across a wide range of configurations and architectures. The accompanying pseudo-code, presented later, outlines the steps for this integration. A key aspect to ensure valid comparative evaluation involves the ordering of various components, such as the stopping gradient and batch size.

---

**Algorithm 1** Multiview training in a PyTorch-like style

---

**for** x in loader **do**

    # Generate Multiviews
    $views \leftarrow \mathbf{T_1}(\mathrm{x}), \mathbf{T_2}(\mathrm{x}), ..., \mathbf{T_n}(\mathrm{x})$
    **for** $v_i$ in views **do**
        $z_i \leftarrow$ model.backbone($v_i$)

        # Keep Same Configuration for each View
        **if** Projection or Prediction Head **then**
            $p_i \leftarrow$ model.projection_head($p_i$)
            $z_i \leftarrow$ model.projection_head($z_i$)
        **end if**
        **if** stop gradient **then**
            $p_i \leftarrow$ model.projection_head($p_i$)
        **end if**
        **if** Others Technologies **then**
            $p_i \leftarrow$ model.Other_Technologies($p_i$)

        **end if**
        $P$.append($p_i$)
        $Z$.append($z_i$)
    **end for**

    # Compute the mean embedding
    $z_{\mathrm{mean}} \leftarrow$ torch.mean($Z$, dim = 0)
    $losses \leftarrow [\text{torch.criteria}(p_j, z_{\mathrm{mean}}) \text{ for } p_j \text{ in } P]$
    $loss \leftarrow$ torch.mean($losses$)
**end for**

---

**2. Reimplementation Details and Code Availability**    To facilitate the reproducibility of our research, we provide a comprehensive description of all experiment-related components, including hyperparameters, dataset selection, data augmentations, and architectural details, in Appendix 9.

We make available two distinct codebases for straightforward reimplementation, both of which can be accessed at **[Anonymous]** https://anonymous.4open.science/r/Multiple-Positive-View-F043/README.md:

1. **Lightly-based Repository:** This repository offers a user-friendly implementation using the Lightly Python framework (Susmelj et al., 2020). It is designed for ease of use and rapid replication of the main experimental results (Benchmark and Efficiency) presented in this manuscript.

2. **AutoSSL Library:** Tailored for researchers seeking granular control and comprehensive diagnostics, we introduce a specialized library named *AutoSSL*. The library is fully con-

figurable via a single configuration file and covers a wide range of architectural and hyperparameter options. Additionally, it supports batch tuning for an extensive set of configurations, enabling ablation studies and in-depth investigation using advanced monitoring tools.

**Additional Metrics for Evaluation** We encourage incorporating the following metrics into your original code for a more comprehensive analysis and diagnosis of the model performance: Standard Deviation (STD) of views, STD of batches, STD of feature representations, K-Nearest Neighbors (KNN) accuracy, Linear classification accuracy, Running time, Forward propagation time, Backward propagation time, dataloader time, GPU utilization, etc.

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

APPENDIX: TABLE OF CONTENTS

# 8    APPENDIX: THEORY BACKGROUND

## 8.1    PREREQUISITE KNOWLEDGE

This paper relies on a set of key mathematical conditions and properties that lay the groundwork for our theoretical and methodological framework. For any random variables $\mathbf{x}, \mathbf{y}$ and $\mathbf{z}$, the key principles are enumerated as follows:

1. **Multivariate Chain Rule:** The chain rule can be extended to encompass cases involving three or more random variables. The formulation of this principle allows for different permutations of the random variables involved, as represented by the following equation:

$$\begin{aligned} I(\mathbf{x}; \mathbf{y}; \mathbf{z}) &= I(\mathbf{y}; \mathbf{z}) - I(\mathbf{y}; \mathbf{z} \mid \mathbf{x}) \\ &= I(\mathbf{x}; \mathbf{y}) - I(\mathbf{x}; \mathbf{y} \mid \mathbf{z}) \\ &= I(\mathbf{x}; \mathbf{z}) - I(\mathbf{x}; \mathbf{z} \mid \mathbf{y}) \end{aligned} \tag{12}$$

2. **Multivariate Mutual Information (Multiple View Case):** For multiple views exceeding two, the multivariate mutual information can be generalized as:

$$I\left(\mathbf{x}_1; \mathbf{x}_2; \ldots; \mathbf{x}_{n+1}\right) = I\left(\mathbf{x}_1; \ldots; \mathbf{x}_n\right) - I\left(\mathbf{x}_1; \ldots; \mathbf{x}_n \mid \mathbf{x}_{n+1}\right). \tag{13}$$

## 8.2    RETHINKING SELF-SUPERVISED LEARNING FROM A HOLISTIC PERSPECTIVE

The quest to define what constitutes a "good representation" has sparked extensive discourse within the domain of machine learning (Shwartz-Ziv & LeCun, 2023; Wang et al., 2022; Shwartz-Ziv et al., 2022). Without a robust and rigorous theoretical foundation, it is challenging to delve deeper into why multiview approaches, or any method, for that matter, can be effectively optimized. A dominant viewpoint advocates for an information-theoretic approach, suggesting that effective representations in supervised learning models can be understood through the lens of the IB principle. Such representations strive to encapsulate the minimally sufficient statistics necessary to satisfy specific conditions.

Nevertheless, the intrinsic complexity of self-supervised learning defies easy categorization or explanation solely based on strict assumptions or monolithic theories. Given this backdrop, we cautiously select a widely-accepted consensus: an effective embedding should incorporate task-specific information.

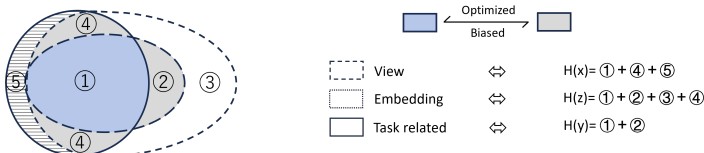

Figure 10: Visual representation of information areas relevant to self-supervised learning

**Interpretation of Visualized Areas Through Mathematical Formulations**    Directly following Figure 10, we aim to elucidate the mathematical foundations behind each visualized area:

- Area 1 (Valid Representation): Corresponds to $I(z; y)$.
- Area 2 (Redundancy Bias [5]): Defined as $I(z; v) - I(z; y) = H(z) - I(z; y)$
- Area 4 (Sufficiency Bias $\alpha$ [4]): Given by $I(v; y) - I(z; y)$
- Area 3 (Redundancy Optimized): Calculated by $H(z) - I(v; y) - I(z; v)$
- Area 5 (Sufficiency Bias $\beta$ [6]): Expressed as $H(y) - I(v; y)$

---

[5]View Specific Bias; Optimizable for each view
[6]View Invisible Bias; Non-optimizable

**Formulaic Information Concept into the Visual Elements**   With the areas now explicitly defined, we can further map them to specific mutual information metrics:

Meaningful Representation:

$$\text{View } v := \text{Area}(1) + \text{Area}(4) + \text{Area}(5)$$
$$\text{Embedding } z := \text{Area}(1) + \text{Area}(2) + \text{Area}(3) + \text{Area}(4)$$
$$\text{Task-Related } y := \text{Area}(1) + \text{Area}(2)$$

Mutual Information:

$$I(v; y) = \text{Area}(1) + \text{Area}(4)$$
$$I(z; y) = \text{Area}(1)$$
$$I(z; v) = \text{Area}(1) + \text{Area}(2)$$

Conditional Mutual Information Metrics:

$$I(v; y \mid z) = \text{Area}(4)$$
$$I(v; z \mid y) = \text{Area}(2)$$
$$I(z; y \mid v) = 0$$

### 8.3   OUR ASSUMPTIONS AND SCOPE OF APPLICATION

Our approach is anchored in the principles of the Information Bottleneck (IB) and mutual information within a Self-Supervised Learning (SSL) framework. It is particularly relevant for most SSL contrastive learning methods, especially those focused on dual-view scenarios. Our method makes minimal specific assumptions, primarily adopting a fundamental representational hypothesis similar to Federici et al. (2020) that $I(z, y|v) = 0$. This suggests a certain level of information redundancy across different views. Each view, therefore, contains unique information vital for understanding the overall dataset, along with overlapping information shared with other views. Additionally, we assume that data augmentation and processing do not significantly alter the relationship of information across views. These assumptions are central to our method's design, influencing our approach to optimizing the objective function and handling embedded representations.

On the other hand, we posit that existing contrastive learning loss functions, exemplified by CPC (Linsker, 1988) and SimCLR (Chen et al., 2020), are analogous to minimizing $I(z_1, z_2)$. The closer a loss function aligns with this objective, the more effectively it can benefit from minimizing and reducing sufficiency bias. Our empirical experiments corroborate this assertion. However, architectures deviating from this goal might experience adverse effects, as observed in DCL. Interestingly, when applying our method in DINO to replace the multi-cropping strategy, we observed comparable results, achieving 79.25% vs. 79.44% on ImageNette and 35.77% vs. 35.84% on Tiny ImageNet. Despite the method not being explicitly designed around $I(z_1, z_2)$, it still demonstrates competitive performance.

### 8.4   PREVIOUS ASSUMPTIONS

Previous methods made many assumptions, and its ultimate goal is to approximate the labeled objective function of the unlabeled pretext task. We present a list to illustrate the potential risks that may arise from improper assumptions.

#### 8.4.1   REPRESENTATION

**Assumptions for Single View**   As postulated in Federici et al. (2020), it is reasonable to assume that the embedding $z$ serves as the representation of the image $v$. Under this assumption, the conditional mutual information between an image $v_i$ and a task $y$ given its embedding $z_i$ becomes zero, mathematically denoted as:

$$I(z_i; y \mid v_i) = 0, \tag{14}$$

indicating that the task-related information remains invariant between the image and its corresponding embedding. It is the sole assumption adopted by our method, which considers the situation for each view separately, showcasing the method's reliability.

**Assumptions across Multiple Views**  When considering multiple views, we naturally extend this to the concept of mutual representation. In this context, the embedding captures partial task-unrelated information. We formalize this weak assumption through the following equation:

$$I(z_i; y \mid v_j) = 0 \tag{15}$$

This is considered a "weak" assumption due to its overly stringent nature. It imposes the constraint that the task-related information between $v_i$ and $y$ becomes zero when conditioned on an embedding $z_j$ from a different view. This stringent condition could potentially limit the model's ability to leverage shared information across multiple views, potentially hindering its performance in multiview scenarios.

### 8.4.2 REDUNDANCY

**Assumptions across Two Views**  The view $v_1$ is considered redundant with respect to $v_2$ for the task $y$ if and only if:

$$I(y; v_1 \mid v_2) = 0 \tag{16}$$

Given $v_2$, $v_1$ fails to offer any additional information pertaining to $y$, thereby rendering it redundant. Furthermore, the concept of mutual redundancy is introduced based on the assumption of representation discussed earlier in Eq.14. Aligning with the findings of Federici et al. (2020), the following equation holds: $I(y; v_1 \mid v_2) = I(y; v_2 \mid v_1)$, which signifies that $v_1$ and $v_2$ are mutually redundant, as either view provides sufficient information about $y$ when the other view is given.

**Assumptions across Multiple Views**  It can be extended to multiple views, if $v_i$ and $v_j$ are mutually redundant, the following equation holds:

$$I(v_i; y \mid v_j) = I(v_j; y \mid v_i) = 0 \quad \text{(mutually)} \tag{17}$$

This signifies that $v_i$ and $v_j$ are interchangeable in the context of providing information about $y$. That is, knowing either $v_i$ or $v_j$ suffices to capture all task-related information, making the other view unnecessary and mutually redundant. However, as Federici et al. (2020) illustrated, the assumption is too strict, the two view scenario cannot be simply extended to this case.

### 8.4.3 MINIMALITY AND SUFFICIENCY

**Assumptions for Single View:**  Inherited from IB theory (Tishby et al., 1999) of supervised learning, two critical assumptions of sufficiency and minimality are commonly delineated:

$$
\begin{aligned}
\text{Sufficiency:} \quad & I(v_i; y \mid z_i) = 0, \\
\text{Minimality:} \quad & I(v_i; z_i \mid y) = 0.
\end{aligned} \tag{18}
$$

In contrastive learning, an indirect formulation frequently arises in the literature (Tian et al., 2020b; Federici et al., 2020) can be seen in the following equation. It should be emphasized that transitioning from Eqs.18 -19 is not trivial. Most of these formulations implicitly depend on representation assumptions, as encapsulated in the previous Eq.14.

$$I(v_i; y) = I(z_i; y) = I(z_i; v_i) \tag{19}$$

**Assumptions across Multiple Views:**  Further, contrastive learning can be extended to maximize mutual information between different view representations, thereby obtaining sufficient representations (Oord et al., 2018; Arora et al., 2019; Tian et al., 2020a; Wang et al., 2022). The fundamental premise is that all views should share and only share the minimal task-related information. This is informed by the understanding that all supervisory information in one view stems from another (Wang et al., 2022).

$$
\begin{aligned}
\text{Sufficiency:} \quad & I(v_i; y \mid z_j) = 0 \\
\text{Minimality:} \quad & I(v_i; z_j \mid y) = 0 \\
or & \\
\text{Mutually Minimal Sufficient:} \quad & I(v_i; y) = I(z_j; y)
\end{aligned} \tag{20}
$$

However, the sharing of unbiased effective task-related information between these views is complex and accompanied by difficulties. Wang et al. (2022) demonstrated that contrastive learning models risk overfitting by sharing information excessively between views. This complexity emerges because each view could harbor unique, non-shared information, complicating the sufficiency and minimality conditions.

### 8.5 PROOFS IN OUR METHOD

#### 8.5.1 LINK BETWEEN REWORKED OBJECTIVE AND INFORMATION BOTTLENECK THEORY

*Proof.*

This is a detailed proof (for Eqs.2 -3), we proposed an objective function that is based on the information bottleneck theory. We aim to highlight the established connection rather than presenting a novel concept.

$$
\begin{aligned}
\min_{p(z|x)} I(V; Z \mid Y) - \beta I(Z; Y) &\overset{12}{=} \min_{p(z|x)} I(V; Z) - I(V; Z; Y) - \beta I(Z; Y) \\
&\overset{12}{=} \min_{p(z|x)} I(V; Z) - I(Z; Y) + I(Z; Y \mid X) - \beta I(Z; Y) \\
&= \min_{p(z|x)} I(V; Z) - (\beta + 1)I(Z; Y) + I(Z; Y \mid X) \\
&then \\
&\overset{14}{=} \min_{p(z|x)} I(V; Z) - (\beta + 1)I(Z; Y) \\
&\leq \min_{p(z|x)} I(V; Z) - \beta I(Z; Y) \\
&or \\
&= \min_{p(z|x)} I(V; Z) - \beta I(Z; Y) + \underbrace{I(Z; Y \mid X) - I(Z; Y)}_{\leq 0} \\
&\leq \min_{p(z|x)} I(V; Z) - \beta I(Z; Y)
\end{aligned} \tag{21}
$$

For given view, $I(V; Y)$ is fixed:

$$
\begin{aligned}
I(V; Y) &= I(Z; Y) + I(V; Y|Z) \\
&\to \min_{p(z|x)} -I(Z; Y) = \min_{p(z|x)} I(V; Y|Z) \\
&Then \\
&\min_{p(z|x)} I(X; Z|Y) - \beta I(Z; Y) \propto \min_{p(z|v)} \underbrace{I(V; Z|Y)}_{\text{Minimality bias}} + \underbrace{\beta I(V; Y|Z) + \sigma}_{\text{Sufficiency bias}}
\end{aligned} \tag{22}
$$

As a result, we build a connection with IB theory in Eq.2. By utilizing the invariant nature of label and view information in optimization, we rewrite the formula into redundancy bias and sufficiency bias in Eq.3.

#### 8.5.2 OBJECTIVE OF CONTRASTIVE LEARNING UNDER STRICT ASSUMPTIONS

This is a detailed proof (for Eqs.4 -6) that examines and expands on what is being optimized by traditional two-view contrastive learning. Without making any assumptions, we observe that the terms $I(z_2; y \mid z_1)$ and $I(z_1; z_2 \mid y)$ are the source of the error. While it is possible to achieve an optimal outcome under certain strict assumptions, these are difficult to satisfy in (most) real cases.

*Proof.*

$$
\begin{aligned}
\text{InfoNCE} &\leq I(z_1; z_2) \\
&\overset{12}{=} I(z_1; z_2 \mid y) + I(z_1; z_2; y) \\
&\overset{12}{=} I(z_1; z_2 \mid y) + I(z_2; y) - I(z_2; y \mid z_1) \\
With\ &Assumptions: \\
&\overset{15}{=} I(z_1; z_2 \mid y) + I(z_2; y) \\
&\overset{20}{=} I(z_2; y) + I(v_2; z_2 \mid y) \\
&\overset{20}{=} I(z_2; y) - \beta I(v_2; z_2 \mid y)
\end{aligned} \tag{23}
$$

*Reform in symmetric Loss function:*

$$\text{InfoNCE} \leq I(z_1; z_2) + I(z_2; z_1)$$

$$= \sum_{n=1}^{2} I(z_i; y) - \beta \sum_{n=1}^{2} I(v_i; z_i \mid y) \quad \text{(where } \beta = -1\text{)}$$

*Remark:* In the derived formula, the term $I(v_2; z_2 \mid y)$ introduces a negative contribution compared to the theoretical objective. This discrepancy can be eliminated under a specific set of assumptions across multiple views, namely, $I(v_1; v_2) = I(v_2; v_3) = I(v_1; v_3) = I(z_i; z_j)$. This condition implies that all embeddings are sharing only task-related information. Under these strict conditions, the term $I(v_i; z_i \mid y)$ can be considered negligible, effectively reducing it to zero, thereby aligning the expression with the theoretical framework.

## 8.6 ADAPTIVE WEIGHT ADJUSTMENT BY CIRCULAR SYMMETRIC LOSS

$$\mathbf{\Omega} = \sum_{i=1}^{n} I(v_i; z_i \mid y) - \sum_{i=1}^{n} \beta I(z_i; y) \tag{24}$$

In fact, due to their mathematical properties and inherent symmetry in many self-supervised tasks, symmetric loss functions are frequently employed. This symmetry could arise from the task setup itself, such as predicting the context from a word in Church (2017) (or vice versa), or predicting the rotation of an image given a rotated version (Gidaris et al., 2018) (or vice versa). Further, symmetric loss functions often stem from the assumption of exchangeability of views in self-supervised learning tasks (Grill et al., 2020; Chen et al., 2020). In many such tasks, there is no inherent order or preference among the views, and they can be interchangeably treated. This symmetry resonates with the maximum entropy principle (Hadsell et al., 2006), advocating that the optimal model preserves all information about the answer found in the input, which aligns seamlessly with our goal.

As illustrated in Figure 11, extending this symmetry to a multiview framework leads to an objective function that summarizes all views. Given the factors of redundancy and sufficiency bias discussed earlier, not all views will perfectly overlap or share task-relevant information uniformly. In this context, the circular symmetry prioritizes the regions of the feature space where more views intersect while down-weighting the regions that are less commonly shared among the views. This mechanism serves as a smoothing factor, adaptively adjusting the weights to achieve an optimized and comprehensive representation of the feature space.

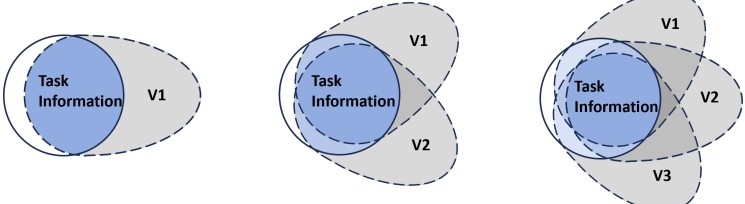

Figure 11: **Visualization of circular symmetry.** In this figure, solid circles represent task-relevant information, while dashed areas indicate task-irrelevant information. Color-coded regions denote the degree of information sharing among multiple views: the blue regions represent task-relevant information, while the grey regions indicate redundancy. The intensity of the color signifies the level of overlap among the views, with deeper hues representing greater overlap.

## 8.7 DIFFERENCE BETWEEN SYMMETRIC AND MULTIVIEW LOSS FUNCTION

Taking the negative cosine similarity as an example. The basic form of the loss is given by:

$$L(q, z) = 1 - \frac{\langle p, z \rangle}{\|p\|_2 \cdot \|z\|_2} \tag{25}$$

When applied to a two-view case, it follows the computation as suggested in BYOL (Grill et al., 2020):

$$L_{v_2} = 0.5 \cdot \left( 1 - \frac{\langle p_0, z_1 \rangle}{\|p_0\|_2 \cdot \|z_1\|_2} + 1 - \frac{\langle p_1, z_0 \rangle}{\|p_1\|_2 \cdot \|z_0\|_2} \right) \tag{26}$$

For the four-view case, we first compute the mean projection embedding $z_{mean}$, then the final loss $L_{v_4}$ for the four-view scenario is then computed by averaging the losses as follows:

$$L_{v_4} = \frac{1}{n} \sum_{i=1}^{n} L(p_i, z_{mean}) \tag{27}$$

Our research indicates the crucial role of Circular Symmetry and Mean Embedding in reducing sufficiency and redundancy biases, thereby bolstering the effectiveness of symmetric loss functions.

## 9 APPENDIX: IMPLEMENTATION DETAILS

As outlined in Statement 7, we designed the benchmarking experiments on various comparative methodologies to be flexibly reimplemented. A comprehensive presentation of datasets, hyperparameters, and other configuration settings is presented to standardize these variables for a fairer comparison. Unless there's an additional statement, these configurations are consistently employed across all experiments. Specific procedures or parameters unique to individual experiments are detailed in Appendix 9.5.

### 9.1 DATASETS

In our experiments, we evaluated five distinct datasets (Krizhevsky et al., 2009; Coates et al., 2011; Howard, 2020; Le & Yang, 2015), each possessing unique characteristics. Table 6 offers an overview of each dataset, detailing the number of classes, total samples, and image size. Of particular note is the STL-10 dataset, which comprises 10 classes, each housing 500 labeled images. This dataset further boasts an additional 100,000 unlabeled images and reserves a separate set of 8,000 images exclusively for independent testing. CIFAR-10 is a standard and balanced dataset with uniform class distribution. CIFAR-100, similar to CIFAR-10, contains more classes, making it suitable to test a model's ability to handle multi-class data. Tiny ImageNet features a larger number of classes and images, which are down-sampled from the original ImageNet (Deng et al., 2009). ImageNette, resembling ImageNet, includes images of varying sizes, highlighting a model's robustness to size and adaptability.

Table 6: **Overview of datasets.**

| Dataset | Samples | Classes | Image Size |
|---|---|---|---|
| CIFAR-10 (Krizhevsky et al., 2009) | 60000 | 10 | 3x32x32 |
| CIFAR-100 (Krizhevsky et al., 2009) | 60000 | 100 | 3x32x32 |
| ImageNette (Howard, 2020) | 13394 | 10 | variable |
| Tiny ImageNet (Le & Yang, 2015) | 110000 | 200 | 3x64x64 |
| STL-10 (Coates et al., 2011) | 113000 | 10 | 3x96x96 |

### 9.2 HYPERPARAMETER OF ARCHITECTURES

In our experiment, we evaluated 8 different SSL methods (Grill et al., 2020; He et al., 2020; Chen et al., 2020; Chen & He, 2021; Bardes et al., 2022a; Yeh et al., 2022; Zhu et al., 2022; Dwibedi et al., 2021). We utilized ResNet-18 (He et al., 2016) with a batch size of 512. The learning rate factor, denoted as *lr_factor*, is determined as the ratio of the *batch size*/128. For optimization, we applied the Stochastic Gradient Descent (SGD) algorithm (Bottou, 2010), incorporating a cosine annealing learning rate schedule (Loshchilov & Hutter, 2017). The learning rate $lr$ within SGD is computed as $lr = 6 \times 10^{-2} \times lr\_factor$.

**BYOL setup** (Grill et al., 2020): In accordance with the BYOL architecture, we implement a projection head consisting of a 2-layer MLP and a prediction head. The projection head takes an input of dimension 512, which expands to a hidden layer of 1024 dimensions. This is followed by batch normalization and a ReLU activation unit, yielding an output dimension of 256. Similarly, the prediction head starts with an input dimension of 256, processes through a 1024-dimensional hidden layer, and outputs a 256-dimensional feature vector.

**MoCo setup** (He et al., 2020): MoCo, sharing structural similarities with BYOL mainly because both leverage a momentum encoder, distinguishes itself by only employing an MLP projection head paired with a memory bank. This projection head transforms the 512-dimensional input into a 512-dimensional hidden layer, undergoes batch normalization and ReLU activation, finally resulting in a 128-dimensional output.

**SimCLR setup** (Chen et al., 2020): Following the setup in SimCLR, it has a projection head identical to MoCo.

**SimSiam setup** (Chen & He, 2021): Consistent with the SimSiam architecture, the projection head comprises three fully-connected layers, each accompanied by Batch Normalization. Importantly, ReLU activations are implemented only after the first two layers, not extending to the final output layer. These layers primarily function at a 2048-dimension space, deviating from the initial layer which accommodates a 512-dimension input. In addition, SimSiam features a 2-layer prediction head. This MLP incorporates batch normalization within its hidden layers and includes a ReLU activation unit, facilitating a dimensional progression of 2048 (input), 512 (hidden), and 2048 (output).

**VICReg setup** (Bardes et al., 2022a): In line with the VICReg architecture, the projection head, analogous to that in BarlowTwins (Zbontar et al., 2021), incorporates a 3-layer MLP. The first layer expands the input from a 512-dimensional space to a 2048-dimensional space. The subsequent layers retain a dimensionality of 2048.

**DCL setup** (Yeh et al., 2022): Following the setup in DCL, it has a projection head identical to MoCo.

**TiCo setup** (Zhu et al., 2022): Following the setup in TiCo, it has a projection head identical to BYOL.

**NNCLR setup** (Dwibedi et al., 2021): In line with the NNCLR architecture, it employs a projection head consisting of three distinct fully-connected layers, with the input, hidden, and output dimensions being 512, 2048, and 256, respectively. Each layer incorporates batch normalization, while ReLU activation is applied following the first and second layers, mirroring the configuration observed in SimSiam. Furthermore, it integrates a 2-layer prediction head, initiating with a dimensional expansion from 256 to 4096, complemented by batch normalization and ReLU activation, culminating in an output dimensionality of 256.

## 9.3 DATA AUGMENTATION

In our experiments, we generate multiple positive views for two and four views utilizing global data augmentation. The underlying data augmentation operations are implemented through `torchvision.transforms` from PyTorch (Paszke et al., 2019). Although each self-supervised learning (SSL) method we employ possesses a distinct data augmentation approach, there is a notable similarity among them. Broadly speaking, the methods utilized can be categorized based on two primary strategies: the SimCLR group and the SimSiam group. While these groups share overarching similarities, they significantly diverge in the selection of parameters in random cropping and color jittering. The SimCLR global augmentation is applied to BYOL, MoCo, SimCLR, VICReg, DCL, TiCo and NNCLR. For SimSiam, its native global augmentation strategy is employed. Below are details of our global augmentation process:

1. Random cropping: The input image is randomly cropped, with the scale ratio of the crop being randomly selected from the minimum scale size to $1$. The crop range for SimCLR is $[0.08, 1]$; for SimSiam, it is $[0.2, 1]$;

2. Horizontal flip: The horizontal flip probability for both global augmentations is $0.5$;

3. Color Jittering: The brightness, contrast, saturation, and hue of the image are individually adjusted by applying a random offset to all pixels in the image uniformly. The alteration

strength is set at $0.5$ for SimCLR, a configuration found to be better suited for datasets with smaller image sizes. In contrast, the SimSiam employs a higher color jittering strength of $1.0$. The probability for both global augmentations is $0.8$ and color jitter for SimCLR is $(0.8, 0.8, 0.8, 0.2)$; for SimSiam $(0.4, 0.4, 0.4, 0.1)$;

4. Grayscale: the probability of grayscale is $0.2$ in both augmentations;

5. Gaussian blur: Gaussian blur is not applied in both SimCLR and SimSiam augmentations (set to $0$ in our experiments);

6. Normalization: Normalization with a mean of $(0.485, 0.456, 0.406)$ and a standard deviation of $(0.229, 0.224, 0.225)$ for both augmentations.

## 9.4 EVALUATION

**KNN Evaluation:** Following the setup in Wu et al. (2018); Caron et al. (2020), we employ the k-Nearest Neighbors (K-NN) approach to assess the feature representations extracted by ResNet-18 (He et al., 2016). The K-NN classifier is trained on the training dataset and evaluated on the corresponding testing dataset. For both the K-NN training and testing datasets, a series of transformations are applied that scale the pixel values to the range $[0.1, 1.0]$ and normalize the images using a mean of $(0.485, 0.456, 0.406)$ and a standard deviation of $(0.229, 0.224, 0.225)$. In our experiment, we set $k = 200$, which means that the class of each test sample is determined by its 200 nearest neighbors (200NN). Furthermore, we utilize a temperature parameter $\tau = 0.1$ to reweight the similarities between the samples during the K-NN classification.

**Linear Evaluation:** Following the methodology of Chen et al. (2020), we freeze the ResNet-18 backbone and train a linear classifier on top using SGD Bottou (2010), with a base learning rate of 0.1 and a momentum of 0.9. The learning rate is adjusted using a cosine schedule Loshchilov & Hutter (2017), and no warm-up phase is adopted. The training data undergoes augmentation with operations such as random resized cropping (224x224) and horizontal flipping, followed by normalization. In contrast, the validation data is resized to 256 pixels, center-cropped to 224 pixels, and similarly normalized.

## 9.5 IMPLEMENTATION DETAILS OF THE EXPERIMENTS

In this section, we provide a thorough explanation of each experiment conducted in our study. Except as specifically noted, all experiments adhere to the same general architecture and augmentation strategies defined above.

### 9.5.1 METHOD: COMPARISON OF MULTIVIEW STRATEGIES

As an extension to the foundational architecture detailed in Section 9.2, Table 1 presents a thorough comparative analysis of various multiview strategies. This evaluation primarily utilizes the K-Nearest Neighbors (KNN) accuracy metric, detailed in Section 9.4. Additionally, computational overheads such as GPU utilization and time complexity, estimated via the Lightning package (Falcon & team, 2019), are reported across all configurations.

The experimental setup specifically utilizes the CIFAR-10 dataset, and the model is trained for 50 epochs using a batch size of 128. To quantify the level of uncertainty in our findings, each strategy is subjected to three independent tests. Both the mean and the standard deviation of the results are reported to provide a comprehensive understanding of the performance of each multiview strategy.

This supplemental section aims to provide detailed insights into the comparative efficiency and effectiveness of different multiview strategies, thereby facilitating a nuanced understanding and smoother replication of our research.

### 9.5.2 EXPERIMENT: TRAIN LOSS, VIEW AND BATCH REGULARIZATION

To provide further clarity, Figure 5 complements the general experimental setup delineated in Sections 9. The presented results are based on experiments conducted over a span of 200 epochs on the ImageNette dataset with a fixed batch size of 512.

We observe consistent patterns across multiple architectures. To illustrate more concretely, we include a plot comparing the training losses between two-view and four-view configurations, with DCL serving as a representative example. Additionally, the figure depicts the variability in standard deviation across views when employing VICReg, it also reports the standard deviation of batch sizes in the context of MoCo.

The purpose of this appendix section is to furnish a comprehensive insight into the nuances of our experimental setup, facilitating a clearer interpretation and easier reproducibility of our results.

### 9.5.3 EXPERIMENT: DOMAIN TRANSFER ASSESSMENT

In the context of self-supervised learning, assessing the generalizability of models across diverse datasets is critical. Complementary to the main text, this section details our efforts in evaluating the domain transfer capabilities of various self-supervised models. The evaluation metric used is linear evaluation, as outlined in Section 9.4.

Using a single Nvidia 4090 GPU, we conduct training for 100 epochs with a batch size of 512, and evaluate the models on out-of-domain datasets. We scrutinize the performance of five key self-supervised models: BYOL (Grill et al., 2020), MoCo (He et al., 2020), SimCLR (Chen et al., 2020), SimSiam (Chen & He, 2021), and VICReg (Bardes et al., 2022a). These models are evaluated over six dataset pairs:

1. STL-10 and CIFAR-100;
2. CIFAR10 and CIFAR-100;
3. ImageNette and CIFAR-10;
4. STL-10 and CIFAR-10;
5. STL-10 and ImageNette;
6. Tiny ImageNet and CIFAR-100.

For each dataset pair, the models are trained on one dataset and their performance is evaluated on the other, facilitating a comprehensive assessment of their generalizability.

The aim of this appendix section is to offer exhaustive details about the domain transfer experiments, thereby assisting in a granular understanding and reproducibility of our assessments.

### 9.5.4 EXPERIMENT: BENCHMARK COMPARISION AND PERFORMANCE

In Table 2 and 3, we assess the KNN accuracy and performance utilizing an Nvidia 4090 GPU with a batch size of 512. The architecture setup follows 9.2. We benchmark the KNN accuracy across 8 methods on 5 datasets in two-view scenarios, conducting training over 800 epochs using Python Lightly (Susmelj et al., 2020). This is extended to four-view scenarios, with training conducted over 200 epochs, applying our method as detailed in Section 7. Evaluation of GPU usage and runtime is conducted with the Pytorch Lightning package. Intermediate results are documented in TensorBoard, with full implementation details available on `https://anonymous.4open.science/r/Multiple-Positive-View-F043/README.md`.

### 9.5.5 ABLATION: BATCH SIZE, VIEW NUMBER, AND DATA AUGMENTATION

In our ablation study using an Nvidia 4090 GPU, we leverage the AutoSSL-library to assess both two-view and four-view scenarios. For the view ablation, we examine the performance of MoCo and VICReg with a batch size of 128 for 50 epochs. In the batch ablation study, we evaluate SimSiam performance using various batch sizes, up to a maximum size of 2048, over a span of 50 epochs. For each architecture used, the setup is detailed in 9.2. For the augmentation ablation, we assess the linear accuracy of SimSiam, utilizing a batch size of 128 across 50 epochs, while employing different augmentation strategies. The data augmentation in the first two parts of the ablation study utilizes a 'global augmentation' strategy, which integrates approaches from both SimCLR and SimSiam, as detailed in section 9.3. Specifically, we employ SimSiam's color jittering and SimCLR's other parameters. For the augmentation ablation part, we align parameters with "global augmentation". However, we either remove or add a specific technique, as required, to analyze its impact.

## 10 APPENDIX: ADDITIONAL ANALYSES

In our experiment, we carefully account for both time efficiency and accuracy, as an exclusive emphasis on either would compromise the fairness of the conclusions. Our aim is to provide valuable insights and pragmatic strategies suitable for future research.

### 10.1 ABLATION: VIEW NUMBER

Multiview contrastive learning techniques like CMC, SwAV, DINO, and VIERegL (Tian et al., 2020a; Caron et al., 2020; 2021; Bardes et al., 2022b) have empirically demonstrated performance improvements with an increased number of views. Nevertheless, these advantages often incur a corresponding rise in computational cost. Our empirical observations, showcased in Figures 7 and 8, validate these trends.

In our experiments with MoCo and VICReg, we noted a consistent increase in both KNN and linear accuracies with the escalation of view counts. This trend validates the use of KNN accuracy as an effective metric for comparative benchmarking. Importantly, our study is the first to identify a trade-off at four views, beyond which accuracy gains diminish as the number of views increases.

Meanwhile, running time correspondingly increases as a by-product of accuracy improvement. We focus on identifying when this cost becomes non-negligible. In MoCo, which utilizes a memory bank, GPU usage peaks at four views and subsequently stabilizes. In the case of VICReg, GPU usage significantly increases utilizing four views and reaches a plateau thereafter. Despite current conventions suggesting that memory banks usually inflate GPU usage, VICReg consumes more computational resources than MoCo. This anomaly is attributed to VICReg's linear computations across batches.

### 10.2 ABLATION: BATCH SIZE

We examined the impact of two-view and four-view configurations on SimSiam's accuracy, with a specific focus on varying batch sizes. Apart from the findings reported in the ablation part, we observed that a four-view setup prevents model collapse in smaller batch sizes, particularly in the range of 52 to 536 where fluctuations in accuracy are significant. As the number of training epochs increases, the influence of batch size on model accuracy becomes more pronounced. This stability is attributed to the additional perspectives offered by multiple views, while "mean embedding" further mitigates unstable gradients in smaller batch training. In the multiview strategy, we also noticed the relationship between the model's running time and batch size exhibits a similar linear trend as with the four-view setup. Consequently, our approach does not compromise the efficiency gains attributable to batch size.

### 10.3 ABLATION: DATA AUGMENTATION

Our work demonstrates the robustness of multiview model performance to diverse data augmentations. While He et al. (2020) initially established the value of data augmentation in enriching feature representations in self-supervised learning, Bordes et al. (2023) extended this understanding by demonstrating that even a single, carefully chosen augmentation can greatly enhance feature richness. However, current research indicates that more augmentation is not always better (Tian et al., 2020b), and not all augmentations are equally effective, as shown in Table 5. Strong augmentations such as grayscale. (Bordes et al., 2023), boosting performance considerably. Excessive augmentation, however, hampers learning and introduces additional errors.

In contrast, our multiview strategy exhibits greater inclusivity in view selection. Our approach captures a wider range of features without requiring carefully chosen augmentation techniques and corresponding parameters, thus offering more valuable prospects for industrial applications.

## 11 APPENDIX: ADDITIONAL INTERMEDIATE DATA

### 11.1 DETAILED RESULTS OF DOMAIN TRANSFER TASKS

The implementation of our domain transfer experiments can be found in Appendix 9.5.3 and the detailed results are summarized in Tables 7 to 9. Notably, our method improves transferability in over half of the evaluated scenarios. In addition, we observe that transferring from ImageNette to STL-10 substantial gains across all five architectures when employing four views, notably for MoCo and VICReg, corresponding to improvements in both 65.79% and 98.76%. Conversely, when transferring from STL-10 to ImageNette, such gains are only observed for BYOL and MoCo. Similar trends are evident across other dataset pairs. This phenomenon may be attributed to dataset-specific characteristics, as described in Section 9.1.

Table 7: **STL-10 and CIFAR-100; CIFAR-10 and CIFAR-100**

| Methods | STL-10 → CIFAR-100 | | CIFAR-10 → CIFAR-100 | |
| --- | --- | --- | --- | --- |
| | Gain | *Vice Versa* | Gain | *Vice Versa* |
| BYOL Grill et al. (2020) | 0.21% | 3.86% | −10.88% | 3.62% |
| MoCo He et al. (2020) | 3.64% | 1.21% | 3.29% | 2.33% |
| SimCLR Chen et al. (2020) | 8.12% | 2.31% | 0.78% | 4.80% |
| SimSiam Chen & He (2021) | 4.61% | −4.12% | −1.11% | 17.24% |
| VICReg Bardes et al. (2022a) | 4.16% | −1.53% | 5.24% | 6.96% |

Table 8: **ImageNette and CIFAR-10; STL-10 and CIFAR-10**

| Methods | ImageNette → CIFAR-10 | | STL-10 → CIFAR-10 | |
| --- | --- | --- | --- | --- |
| | Gain | *Vice Versa* | Gain | *Vice Versa* |
| BYOL Grill et al. (2020) | 2.52% | −3.09% | 0.52% | −0.75% |
| MoCo He et al. (2020) | 60.77% | 5.72% | 0.25% | 7.70% |
| SimCLR Chen et al. (2020) | −1.90% | −2.57% | 2.63% | 1.53% |
| SimSiam Chen & He (2021) | 7.40% | −1.26% | −3.44% | −0.55% |
| VICReg Bardes et al. (2022a) | 32.19% | −1.63% | 1.59% | −8.64% |

Table 9: **STL-10 and ImageNette; Tiny ImageNet and CIFAR-100**

| Methods | STL-10 → ImageNette | | Tiny ImageNet → CIFAR100 | |
| --- | --- | --- | --- | --- |
| | Gain | *Vice Versa* | Gain | *Vice Versa* |
| BYOL Grill et al. (2020) | 2.98% | 2.59% | 1.75% | 3.87% |
| MoCo He et al. (2020) | 0.13% | 98.76% | 2.11% | −1.36% |
| SimCLR Chen et al. (2020) | −0.55% | 7.92% | 6.39% | 1.37% |
| SimSiam Chen & He (2021) | −7.38% | 3.40% | 22.92% | −5.66% |
| VICReg Bardes et al. (2022a) | −2.89% | 65.79% | 8.23% | 7.17% |

### 11.2 DETAILED RESULT OF 800 EPOCHS TRAINING OVER 5 DATASETS

The implementation of our benchmark experiments can be found in Appendix 9.5.4. Table 3 showcases the advantages of employing a four-view in enhancing model performance, characterized by improvements in KNN accuracy and time efficiency. Here, "time efficiency" signifies the reduction in time required to attain 90% and 95% of the final KNN accuracy in a two-view scenario within an 800-epoch training cycle, as facilitated by the adoption of four views. The noticeable variations in the KNN accuracy and time efficiency metrics across all datasets hint at the differential benefits of the four-view approach.

For instance, when focusing on the STL-10 dataset (Coates et al., 2011), a 200-epoch training employing four views allowed six out of eight evaluated methods to surpass the KNN accuracy achieved through 800 epochs of two-view training. However, this enhancement was not observed on CIFAR-10 (Krizhevsky et al., 2009). Noteworthy is the role of the four-view configuration in preventing the collapse of VICReg (Bardes et al., 2022a) on multiple datasets including STL-10, Tiny ImageNet, and ImageNette. (Coates et al., 2011; Le & Yang, 2015; Howard, 2020). In these cases, the implementation of four views serves as a stabilizing factor, in addition to enhancing KNN accuracy.

Regarding time efficiency, the implementation of a four-view configuration can potentially accelerate the training process to reach a comparable level of accuracy, albeit the results showcase notable variance across different methods and datasets. For example, the STL-10 and CIFAR-100 datasets reveal substantial benefits when applying four-view approaches. Contrarily, half of the methods show a slowdown in the ImageNette dataset when adopting the four-view strategy.

Table 10: **Benchmark comparison details:** KNN accuracy and time efficiency for two-view and four-view scenarios.

| Method | Dataset | Accuracy Evaluation | | | | 90% Evaluation | | | | 95% Evaluation | | | |
|---|---|---|---|---|---|---|---|---|---|---|---|---|---|
| | | 200e(2v) | 200e(4v) | Enh. | 800e(2v) | Acc. | Time(2v) | Time(4v) | Speed | Acc. | Time(2v) | Time(4v) | Speed |
| **BYOL** | STL-10 | 69.1% | 84.8% | 22.7% | 81.2% | 73.1% | 17132 | 2320 | 7.4 | 77.2% | 25953 | 4644 | 5.6 |
| | CIFAR10 | 81.8% | 89.2% | 9.0% | 91.3% | 82.2% | 7146 | 3925 | 1.8 | 86.7% | 16230 | 6423 | 2.5 |
| | CIFAR100 | 31.7% | 55.8% | 76.0% | 47.5% | 42.7% | 8171 | 907 | 9.0 | 45.1% | 9213 | 1137 | 8.1 |
| | Tiny ImageNet | 25.0% | 38.3% | 53.2% | 36.3% | 32.7% | 54488 | 5401 | 10.1 | 34.5% | 57797 | 8483 | 6.8 |
| | ImageNette | 81.1% | 80.5% | -0.7%↓[4] | 83.4% | 75.1% | 1452 | 1868 | 0.8 | 79.2% | 1938 | 2708 | 0.7 |
| | **Average** | 57.7% | 69.7% | **32.0%** | 67.9% | 61.2% | 17678 | 2884 | **5.8** | 64.6% | 22226 | 4679 | **4.7** |
| | **STD** | 24.5 | 19.5 | **28.5** | 21.8 | 19.6 | 19077 | 1592 | **3.8** | 20.7 | 19473 | 2605 | **2.7** |
| **MoCo** | STL-10 | 71.3% | 81.5% | 14.3% | 83.5% | 75.1% | 20559 | 6115 | 3.4 | 79.3% | 26413 | 11071 | 2.4 |
| | CIFAR10 | 80.0% | 84.6% | 5.7% | 89.9% | 80.9% | 9867 | 5004 | 2.0 | 85.4% | 20611 | 13762 | 1.5 |
| | CIFAR100 | 39.3% | 54.2% | 37.9% | 56.6% | 50.9% | 7920 | 3263 | 2.4 | 53.8% | 9333 | 4195 | 2.2 |
| | Tiny ImageNet | 29.8% | 35.0% | 17.4% | 41.2% | 37.1% | 51449 | 49235 | 1.0 | 39.2% | 55510 | 78085 | 0.7 |
| | ImageNette | 79.5% | 80.5% | 1.3% | 83.6% | 75.3% | 1294 | 1799 | 0.7 | 79.5% | 2243 | 2462 | 0.9 |
| | **Average** | 60.0% | 67.2% | **15.3%** | 71.0% | 63.9% | 18226 | 13083 | **1.9** | 67.4% | 22822 | 21915 | **1.5** |
| | **STD** | 21.2 | 19.4 | **12.7** | 18.8 | 16.9 | 17732 | 18135 | **1.0** | 17.8 | 18394 | 28395 | **0.7** |
| **SimCLR** | STL-10 | 63.5% | 82.2% | 29.5% | 80.2% | 72.1% | 18617 | 4529 | 4.1 | 76.1% | 23039 | 8396 | 2.7 |
| | CIFAR10 | 75.7% | 83.4% | 10.2% | 88.9% | 80.0% | 14572 | 2966 | 4.9 | 84.4% | 20255 | 15725 | 1.3 |
| | CIFAR100 | 31.8% | 56.3% | 77.0% | 51.9% | 46.7% | 8441 | 1950 | 4.3 | 49.3% | 9245 | 2594 | 3.6 |
| | Tiny ImageNet | 22.3% | 30.4% | 36.3% | 37.8% | 34.0% | 44114 | 58503 | 0.8 | 35.9% | 47406 | 67702 | 0.7 |
| | ImageNette | 78.8% | 84.7% | 7.5% | 86.0% | 77.4% | 1867 | 1178 | 1.6 | 81.7% | 3849 | 1935 | 2.0 |
| | **Average** | 54.4% | 67.4% | **32.1%** | 69.0% | 62.1% | 17522 | 13825 | **3.1** | 65.5% | 20759 | 19270 | **2.1** |
| | **STD** | 23.1 | 21.3 | **25.0** | 20.4 | 18.3 | 14453 | 22366 | **1.6** | 19.3 | 15059 | 24718 | **1.0** |
| **SimSiam** | STL-10 | 56.4% | 82.3% | 45.9% | 70.7% | 63.6% | 18778 | 1004 | 18.7 | 67.1% | 22138 | 1672 | 13.2 |
| | CIFAR10 | 75.2% | 85.6% | 13.8% | 90.1% | 81.1% | 11806 | 3882 | 3.0 | 85.6% | 17788 | 12252 | 1.5 |
| | CIFAR100 | 20.4% | 55.1% | 170.1% | 35.2% | 31.7% | 13099 | 844 | 15.5 | 33.4% | 13962 | 844 | 16.5 |
| | Tiny ImageNet | 14.4% | 30.3% | 110.4% | 28.0% | 25.2% | 49091 | 6748 | 7.3 | 26.6% | 51226 | 10120 | 5.1 |
| | ImageNette | 74.9% | 76.1% | 1.6% | 83.1% | 74.8% | 1817 | 2195 | 0.8 | 79.0% | 2819 | *0*[5] | *0.0*[5] |
| | **Average** | 48.3% | 65.9% | **68.4%** | 61.4% | 55.3% | 18918 | 2934 | **9.1** | 58.3% | 21587 | 4977 | **7.3** |
| | **STD** | 26.2 | 20.7 | **63.3** | 25.2 | 22.7 | 16045 | 2194 | **7.0** | 24.0 | 16146 | 5140 | **6.5** |

[4] Negative accuracy improvement for 200 epochs training.
[5] **italicized number:** Four-view scenario with 200 epochs training doesn't get this accuracy.

Table 11: **Benchmark Comparison details:** KNN accuracy and time efficiency for two-view and four-view scenarios.

| Method | Dataset | Accuracy Evaluation | | | | 90% Evaluation | | | | 95% Evaluation | | | |
|---|---|---|---|---|---|---|---|---|---|---|---|---|---|
| | | 200e(2v) | 200e(4v) | Enh. | 800e(2v) | Acc. | Time(2v) | Time(4v) | Speed | Acc. | Time(2v) | Time(4v) | Speed |
| **VICReg** | STL–10 | ǂ[6] | 68.7% | (45.9%)[7] | ǂ[6] | - | - | - | (18.7)[7] | - | - | - | (13.2)[7] |
| | CIFAR10 | 63.6% | 66.7% | 4.9% | 71.1% | 64.0% | 2484 | 2712 | 0.9 | 67.5% | 3886 | 4623 | 0.8 |
| | CIFAR100 | 31.5% | 28.8% | -8.6%↓ | 39.1% | 35.2% | 3840 | 0[5] | 0.0[5] | 37.2% | 5142 | 0[5] | 0.0[5] |
| | Tiny ImageNet | ǂ[6] | 29.5% | (110.4%)[7] | ǂ[6] | - | - | - | (10.1)[7] | - | - | - | (6.8)[7] |
| | ImageNette | ǂ[6] | 69.8% | (8.9%)[7] | ǂ[6] | - | - | - | (2.1)[7] | - | - | - | (2.3)[7] |
| | **Average** | 23.3% | 52.7% | **32.3%** | 26.3% | 23.6% | 1265 | 542 | **6.4** | 25.0% | 1805 | 925 | **4.6** |
| | **STD** | 22.6 | 19.3 | **43.0** | 25.9 | 23.3 | 1607 | 1085 | **7.1** | 24.6 | 2247 | 1849 | **4.9** |
| **DCL** | STL–10 | 66.7% | 77.4% | 16.0% | 80.7% | 72.6% | 18267 | 7944 | 2.3 | 76.6% | 23063 | 11590 | 2.0 |
| | CIFAR10 | 76.1% | 78.7% | 3.4% | 87.5% | 78.7% | 11501 | 10900 | 1.1 | 83.1% | 18363 | 31746 | 0.6 |
| | CIFAR100 | 43.3% | 54.7% | 26.3% | 56.2% | 50.6% | 7044 | 2896 | 2.4 | 53.4% | 8348 | 3519 | 2.4 |
| | Tiny ImageNet | 29.9% | 30.9% | 3.3% | 40.1% | 36.1% | 43612 | 89042 | 0.5 | 38.1% | 47697 | 107502 | 0.4 |
| | ImageNette | 76.4% | 78.5% | 2.7% | 83.1% | 74.8% | 1594 | 1970 | 0.8 | 78.9% | 3135 | 0 | 0.0 |
| | **Average** | 58.5% | 64.0% | **10.4%** | 69.5% | 62.6% | 16403 | 22550 | **1.4** | 66.0% | 20121 | 30871 | **1.1** |
| | **STD** | 18.7 | 18.9 | **9.4** | 18.3 | 16.5 | 14661 | 33406 | **0.8** | 17.4 | 15487 | 39864 | **0.9** |
| **TiCo** | STL–10 | 68.2% | 82.6% | 21.1% | 80.6% | 72.6% | 13163 | 4215 | 3.1 | 76.6% | 17173 | 8425 | 2.0 |
| | CIFAR10 | 66.7% | 74.8% | 12.1% | 79.5% | 71.5% | 5109 | 1768 | 2.9 | 75.5% | 6991 | 4042 | 1.7 |
| | CIFAR100 | 40.1% | 56.0% | 39.7% | 56.3% | 50.6% | 7362 | 2712 | 2.7 | 53.4% | 8363 | 3390 | 2.5 |
| | Tiny ImageNet | 29.0% | 33.3% | 14.8% | 41.9% | 37.7% | 28403 | 72441 | 0.4 | 39.8% | 30950 | 83700 | 0.4 |
| | ImageNette | 76.3% | 80.8% | 5.9% | 81.5% | 73.3% | 1194 | 935 | 1.3 | 77.4% | 1803 | 1217 | 1.5 |
| | **Average** | 56.1% | 65.5% | **18.7%** | 68.0% | 61.2% | 11046 | 16414 | **2.1** | 64.6% | 13056 | 20154 | **1.6** |
| | **STD** | 18.2 | 18.7 | **11.5** | 16.1 | 14.5 | 9504 | 28034 | **1.1** | 15.3 | 10223 | 31858 | **0.7** |
| **NNCLR** | STL–10 | 58.2% | 76.9% | 32.1% | 77.9% | 70.1% | 21310 | 6727 | 3.2 | 74.0% | 23790 | 9290 | 2.6 |
| | CIFAR10 | 70.7% | 79.3% | 12.2% | 89.1% | 80.2% | 16745 | 8683 | 1.9 | 84.6% | 20151 | 24288 | 0.8 |
| | CIFAR100 | 21.0% | 36.0% | 71.4% | 37.2% | 33.5% | 8414 | 3125 | 2.7 | 35.3% | 8918 | 3426 | 2.6 |
| | Tiny ImageNet | 12.8% | 19.6% | 53.1% | 24.4% | 21.9% | 48246 | 40028 | 1.2 | 23.2% | 48934 | 53121 | 0.9 |
| | ImageNette | 76.5% | 83.3% | 8.9% | 85.8% | 77.3% | 2868 | 1348 | 2.1 | 81.5% | 5111 | 2191 | 2.3 |
| | **Average** | 47.8% | 59.0% | **35.5%** | 62.9% | 56.6% | 19516 | 11982 | **2.2** | 59.7% | 21381 | 18463 | **1.8** |
| | **STD** | 26.1 | 26.1 | **23.9** | 26.8 | 24.1 | 15727 | 14259 | **0.7** | 25.4 | 15404 | 19027 | **0.8** |

[6] ǂ means model collapse
[7] **estimated valued:** Choose the best performance of architectures on the dataset as the performance.

