# OpenReview forum: "Multiple Positive Views in Self-Supervised Learning"
_ICLR.cc/2024/Conference — ICLR 2024 Conference Withdrawn Submission_

### Official Review · Reviewer_pKGj · 2023-10-31

**Soundness:** 3 good
**Presentation:** 2 fair
**Contribution:** 4 excellent
**Rating:** 6
**Confidence:** 3

**Summary:**

This work analyzes the limitations of contrastive learning with two-views and extend it to multiple-views through the lens of information theory. They provide theory to show their objective is a lower bound of the information bottleneck. Finally, their experiments show improved performance and speed-up.

**Strengths:**

The problem is underexplored and the novelty of the paper is strong.

The theoretical analysis is rigorous with clear definitions.

**Weaknesses:**

The presentation can be improved.

In method section,
1) The assumptions of the proposed method are not clearly stated.
2) What are the failure cases for the method?
3) What is an intuitive example of view-invisible bias? Could you please clarify "a mutually exclusive state of Z owing to the invariant nature of label and view information in optimization"? What are the "certain approaches" that " assume sharable task information ..."?  Are there experiments to support eq 10? Adding concrete examples could help to better digest the theorems.

In experiment section,

4) Could you clarify why "we adopted unselected data augmentations that slightly deviate from a Sweet Spot"? Does it affect the improvements over baselines?
5) Is the linear accuracy of the main experiment, table 2&3, omitted?
6) Is the method scalable to medium size datasets such as ImageNet?

Ablation study is not a key contributions to the SSL.

For three variables MI is defined as I(x; y, z) = I(y; z) - I(y; z | x). There are typos in eq 12.

**Questions:**

This is an interesting paper in multiple aspects, however the presentation can be significantly improved.

---

> ### Author Response · Authors · 2023-11-19
> **Reply to reviewer pKGj (part1)**
>
> Dear Reviewer,
> Thank you for your insightful comments. Your suggestions have helped us greatly improve the manuscript, and the corresponding modified parts are underlined in green.
>
> ### Response to Q1
>
> It is indeed essential to clarify the assumptions of our method. As mentioned in section 3.2.1 of our paper, SSL inherently implies a series of assumptions that can be established only under an exceedingly strict set of conditions. These conditions act as limiting constraints on stability and consistency. Thus, our work primarily focuses on discarding rather than defining more assumptions. We want to draw the reader's attention to this aspect, and our discussion of assumptions unfolds in three stages. Additionally, our assumptions are minimal, with the Information Bottleneck being the primary one. We believe the appropriately revised article now offers a reasonable exposition of these assumptions.
>
> **1. Clarifying Theoretical and Hypothetical Context**:
> As detailed in the first paragraph of the method on Page 3, we state the theory and assumption before introducing the method.
>
> **2. Concretizing Inferential Assumptions**:
> In Section 3.1, third paragraph, we explicitly add the necessary representative assumptions for formula derivation.
>
> **3. Discussing Limitations of Assumptions and Comparing with Previous Assumptions**:
> In Section 3.22, the second paragraph, we compare and illustrate the assumptions we have discarded. Lastly, as this content is not the focal point of our work, we elaborate on the assumptions in Appendix 8.3, specifically emphasizing the application scope and limitations within this assumption framework.
>
> We believe this will provide a valuable reference for readers interested in understanding the underpinnings of our approach.
>
> ### Response to Q2
>
> In our paper, we mentioned "DCL perform poorly on a smaller dataset are highly probable to fail on larger datasets as well. Theoretically, methods that strive to maximize MI may be more successful". Since failure cases are rare and space constraints, and Max-->MI is almost consensus in most methods, we only briefly mention that in the main text.
>
> However, your insightful comment has prompted us to reconsider your 1st question. So far, we have observed positive effects across nearly all datasets and methods, yet DCL[1] and Barlow Twins[2] failed. The key to DCL's approach is replacing the original objective function with a ratio of similarities between positive and negative samples, eliminating the negative-positive-coupling (NPC) effect. On the other hand, Barlow Twins focuses on preventing collapse through the cross-correlation matrix. These methods share one commonality: they diverge significantly from information entropy.
>
> The discussion of these failure cases, as examples outside our assumption scope, is intrinsically linked to our assumptions about our method (See updated Appendix 8.3). We are very grateful for your insight, prompting this connection.
>
> ### Response to Q3
>
> >What is an intuitive example of view-invisible bias?
>
> "View-invisible bias" refers to the phenomenon where crucial information for a downstream task, such as prior knowledge in domain A classification, is missing in the enhanced views (e.g., cropped images). The essential information needed for classification is represented by the solid-line circle in the figure 2.
> When we apply enhancements like cropping to a view, there's a high probability that necessary information for classification will be lost. Different views may crop different areas, resulting in non-overlapping missing information.
> Therefore, in scenarios with a limited number of views (e.g., 2), the amount of lost information increases. Having more views can mitigate this issue. However, the information contained in view 1 alone can only cover a part of the necessary information for the task. This bias demonstrates the importance of considering multiple views
>
>
>
> >Could you please clarify "a mutually exclusive state of Z owing to the invariant nature of the label and view information in optimization"?
>
> Good question.
>
> In eq 2-3,
>
> $\min _{p(z \mid v)} I(V ; Z \mid Y)-\beta I(Z ; Y)+\sigma$
>
> $\stackrel{21}{=} \min _{p(z | v)} I(V ; Z)-(\beta+1) I(Z ; Y)+\sigma$
>
> $\stackrel{22}{\propto} \min \_{p(z \mid v)} \underbrace{I(V ; Z \mid Y)}\_{\text {Minimality bias }}+\underbrace{\beta I(V ; Y \mid Z)+\sigma}\_{\text {Sufficiency bias }}$
>
> You might be curious about why I(Z;Y) can be proportionally equivalent to I(V;Y|Z). This is due to the fact that I(V;Y) = I(Z;Y) + I(V;'Y|Z). During a certain step in the gradient optimization process, the information from both V and Y remains constant. As a result, these two terms are mutually exclusive and both fall within the part that can be optimized by Embedding Z. We have made corresponding modifications in the text to ensure a clear exposition of this concept.

---

> ### Author Response · Authors · 2023-11-19
> **Reply to reviewer pkGj (part2)**
>
> >What are the "certain approaches" that " assume sharable task information ..."?
>
> We rewrite it as "...prior approaches focused on minimizing $I(z_1, z_2)$ assume that the two views share solely task-relevant information and are independent of redundancy information,"
>
> >Are there experiments to support eq 10? Adding concrete examples could help to better digest the theorems.
>
> To address your question about experiments supporting equation 10, we draw on the intuition that the information shared among variables cannot exceed the total amount of information contained in those variables combined. This is akin to the concept of meaning embeddings distilling redundant information. A visual representation from MacKay's "Information Theory, Inference, and Learning Algorithms" illustrates this point:
> ```
> =====================H(X,Y)======================
> ===============H(X)============|
>                     |============H(Y)============
> =======H(X|Y)=======|====I(X;Y)===|====H(Y|X)====
> ```
>
> However, it's important to note that this visualization is an analogy and not an exact representation. To empirically validate equation 10, we conducted statistical tests on each model. Specifically, we employed binomial statistics tests. The detailed methodology and results can be found on our anonymous GitHub page: https://anonymous.4open.science/r/Multiple-Positive-View-F043/Analysis/equation10.ipynb
>
> As for the results:
> - Number of samples satisfying the condition: 391 out of 400
> - Proportion of satisfactory samples: 0.98
> - Binomial test p-value for threshold 0.96: 0.0403
>
> These findings imply that the proportion of samples satisfying $I(z_1;\ldots;z_n)\leq H(z\_{\text{mean}})\leq{H(z_1,\ldots,z_n)}$ is significantly higher than a set threshold (e.g., 85%). This provides strong evidence that most of your data follows the relationship proposed in equation 10.
>
>
> ### Response to Q4
>
> We adopt unselected data augmentations that slightly deviate from a Sweet Spot[3] made to demonstrate the robustness of our method, and it does not significantly impact the improvements over baselines.
>
> As discussed in Tian et al., 2020b[3], there is a consensus that the relationship between data augmentation and training effectiveness follows a U-shaped curve. Carefully selected augmentations can minimize view-invisible bias and view-specific bias, achieving an optimal solution for \( I(Z1, Z2) \) as the objective function in ideal scenarios.
>
> However, we ensured the use of a broadly accepted augmentation strategy, recognizing that perfect augmentation does not exist and is inherently random. Secondly, if we designed specific augmentations for each model, it would be challenging to compare improvements across different baselines due to our strategy or the augmentation. Thirdly, according to our theory, nearly any point on the U-shaped curve can be leveraged to eliminate these biases through multi-view strategies. Our experimental results validate that our approach is capable of challenging well-selected baselines and augmentations.
>
> ### Response to Q5
> Both linear probing and K-Nearest Neighbors (KNN) are common evaluation metrics in the SSL framework, and their results are often highly correlated. However, in scenarios where class decision boundaries may not be linear, we find KNN to be more persuasive due to its robustness to intra-class variations and a more interpretable feature space.
>
> Additionally, our intention is to align our metrics with established benchmarks like those outlined in the [Benchmarks — lightly 1.4.20 documentation](https://docs.lightly.ai/self-supervised-learning/getting_started/benchmarks.html#cifar-10). This alignment facilitates easier comparison and consistency with prevailing standards in the field.

---

> > ### Author Response · Authors · 2023-11-19
> > **Reply to reviewer pkGj (part3)**
> >
> > ### Response to Q6
> >
> >
> > Currently, our experimental configurations include:
> > 1. Small data volume, low resolution
> > 2. Large batch data, low resolution
> > 3. Small batch data, high resolution
> >
> > Across these varied scenarios, we have consistently observed significant improvements due to our strategy. Indeed, we are in the process of conducting further experiments for our next phase of research, which aligns with our experiences in settings 1-3, as shown in the following table:
> >
> > | Method  | Backbone | Batch | Epochs | Linear-2V | Linear-4V |
> > | ------- | -------- | ----- | ------ | --------- | --------- |
> > | SimCLR* | Res50    | 256   | 100    | 63.2      | 70.5      |
> > | DCL     | Res50    | 256   | 100    | 65.1      | 64.7      |
> > | VICReg  | Res50    | 256   | 100    | 63.0      | 66.4      |
> > | BYOL    | Res50    | 256   | 100    | 62.4      | 68.5      |
> >
> > We acknowledge there are still questions regarding bigger scale experiments. These involve more time-consuming and labor-intensive efforts, potentially extending beyond the scope of this paper.
> >
> > ### Response to Other
> > While ablation studies are not the main contributions to the SSL, our focus remains on providing theoretical support and extensive experimental evidence for the efficacy of multi-view **strategies** as components. Researchers may need to know how to use it and its effects.
> > Regarding your comment on equation 12, we appreciate the notice of typos. The correct expression for three-variable mutual information is indeed \( I(x; y, z) = I(y; z) - I(y; z | x) \). This will be amended to reflect accurately in our work.
> >
> >
> > Reference:
> > [1] Yeh et al., 2022 "Decoupled contrastive learning."
> >
> > [2] Zbontar et al., 2021 "Barlow twins: Self-supervised learning via redundancy reduction."
> >
> > [3] Tian et al., 2020b "What makes for good views for contrastive learning?"

---

### Official Review · Reviewer_K5DW · 2023-10-31

**Soundness:** 2 fair
**Presentation:** 3 good
**Contribution:** 2 fair
**Rating:** 5
**Confidence:** 4

**Summary:**

The paper considers using multiple positive views in training the SSL model. The paper derives certain theoretical aspects and advantages of introducing the multi view self-supervised learning method. Empirical evidence demonstrates the superiority of the proposed method.

**Strengths:**

S1. The paper explores the integration of multiple positive views during the training of the SSL model.

S2. It establishes specific theoretical implications and benefits associated with the implementation of the multi-view self-supervised learning approach.

S3. Empirical findings affirm the effectiveness of the proposed methodology.

**Weaknesses:**

W1. The paper lacks novelty. There are many works introducing multiple views in the SSL pre-training, including DINO, SwAV  and so on, The paper does not compare with these well known SOTA methods.

W2: It is unclear what is the explicit loss function of the proposed method (although it seems Eq. (11) is the loss), and how it hears advantages over SOTA or how it distinguishes in motivation between the existing multiview method such as SwAV (clustering based method with multi-views).

W3: It is unclear if there are fairness issues during the training (empirical evidence), i.e., does the proposed multi-view contrastive learning simply benefits from more "effective epochs" because of its multi-view training (more data in each batch) in comparison to other SOTA methods?

Please help clarify the above concerns.

[A] Mathilde Caron et al., SwAV: Unsupervised Learning of Visual Features by Contrasting Cluster Assignments.

[B] Mathilde Caron et al., . Emerging Properties in Self-Supervised Vision Transformers

**Questions:**

Please see the above weakness for the questions to be addressed. Please correct me during rebuttal, if there is any misunderstanding.

---

> ### Author Response · Authors · 2023-11-19
> **Reply to Reviewer K5DW  (Part1)**
>
> Dear Reviewer,
>
> We appreciate your concerns regarding SOTA models and their multiview strategy. It is a critical perspective to understand our contribution. The updated part has been underlined green.
>
> So in addressing the Reviewer's comment, we aim to elucidate the unique aspects of our approach in the context of SSL (Self-Supervised Learning) architectures. Our work diverges from methodologies like DINO[1] and SwAV[2], which adopt a multi-cropping strategy, as it does not propose a new architectural baseline. Instead, it focuses on integrating a multiview strategy, an effective and integral component, within the framework of contrastive learning. Our research is more inclined towards functionality, addressing an unexplored area in current research.
>
> Particularly, upon reviewing existing methods, we observe a notable lack of discourse, both theoretical and empirical, regarding such multi-view strategies. Our research aims to strike a balance between achieving state-of-the-art (SOTA) performance and enhancing the robustness, generalizability, and interpretability of the multiview strategy. The balance is essential in advancing the field, and our work contributes significantly to this objective.
>
>
> ### Response to W1, Q1
>
> It's indispensable to compare with state-of-the-art (SOTA) architectures. Accordingly, we have updated, reiterated, and supplemented our data to more clearly demonstrate the efficacy of our component:
> 1. **Directly Compare:** We compared our method with multi-cropping and other strategies, observing a clear advantage in our approach (refer to Table 1 for details).
> 2. **Change Strategy on Dino and SwAV**: Applying our strategy to these methods demonstrated accuracy comparable to multi-cropping, though this is likely coincidental (for the underlying reasons, please refer to Appendix 8.3).
> 3. **Enhanced Baseline vs.  (DINO, SwAV)**: Comparing our improved baseline with DINO and SwAV,  we noted that methods enhanced by our strategy generally match even higher than these architectures (please see Table 2 for details)
>
> ### Response to W2
>
> In our paper, to clarify and compare loss function, we follow the notations of existing methods. Since our multi-view strategy is a component for assembling multiple views, it does not involve specific metrics (e.g., cosine similarity). For instance,
>
> In SwAV[2]:
> $$L\left(\mathbf{z}\_{t_1}, \mathbf{z}\_{t_2}, \ldots, \mathbf{z}\_{t_{V+2}}\right)=\sum_\{i \in\{1,2\}} \sum_\{v=1}^{V+2} \mathbf{1}_\{v \neq i} \ell\left(\mathbf{z}\_{t_v}, \mathbf{q}\_{t_i}\right)$$
>
> In CMC[3]:
> $$\mathcal{L}\_F=\sum\_{1 \leq i<j \leq M} \mathcal{L}\left(V_i, V_j\right)$$
>
> In our Paper,
> $$L\left(\mathbf{z}\_{1}, \mathbf{z}\_{2}, \ldots, \mathbf{z}\_{n}\right)=\sum\_{i=1}^{n} \ell\left(\mathbf{z}\_{mean}, \mathbf{z}\_{i}\right) \; \;\; \text{where} \; \; z_{mean}=\sum\_{i=1}^{n}z_{i}/n $$
> Furthermore, as illustrated in Table 1, we provide a comprehensive summary of nearly all existing multiview methods, including SwAV. This table facilitates a comparison of their accuracy, loss functions, and underlying design, supplemented by simple illustrations.
> We wish to emphasize that our method primarily derives its motivation from critically reviewing the inherent biases in existing contrastive learning approaches(e.g. InfoMin[4]), leading us to design a multiview strategy aimed at mitigating these biases. In contrast, multi-cropping largely draws inspiration from blending local and global perspectives, which is a more intuitive approach. While this strategy is employed in DINO and SwAV, as expressed in your W3 concerns, our paper addresses numerous multiview-specific questions like “Does the proposed multiview contrastive learning simply benefit from more 'effective epochs'?”. These discussions are crucial for the SSL community, contributing to the advancement of model interpretability and robustness.

---

> > ### Author Response · Authors · 2023-11-19
> > **Reply to Reviewer K5DW (Part2)**
> >
> > ### Response to W3: Addressing Fairness Concerns
> >
> > We place high importance on fairness in our experiments,  we have updated them to include three key aspects:
> >
> > 1. **Equal Training Epochs**: We ensured an equal number of training epochs across all experiments, demonstrating an improvement in convergence accuracy as shown in Table 2.
> > 2. **Equal Data Exposure with the Same Number of Training Views**: This setup ensures fairness and eliminates the possibility of benefits arising merely from increased view exposure. The updated discussion and results are presented in Table 3.
> > 3. **Time to Reach Specific Convergence Accuracy**: This metric underscores the practical significance of our method. Mirroring the second experiment, it fairly shows our approach's relative effectiveness, as documented in Table 3.
> >
> > Collectively, these experiments address the issue of fairness, offering clear evidence of our method's efficacy under equal conditions.
> >
> > Reference:
> >
> > [1] Caron et al., 2021 "Emerging properties in self-supervised vision transformers."
> >
> > [2] Caron et al., 2020 "Unsupervised learning of visual features by contrasting cluster assignments."
> >
> > [3] Tian et al., 2020a "Contrastive multiview coding."
> >
> > [4] Tian et al., 2020b "What makes for good views for contrastive learning?"

---

> > > ### Comment · Reviewer_K5DW · 2023-11-22
> > > **Thanks for your response**
> > >
> > > Thanks for the response, which somehow addressed some of my concerns. However, I am still confused of explicit loss function used in the paper, as $\ell$ is not properly defined. The novelty is also limited without comparison with other multi-view methods such as SwAV (e.g., Table 2, 3). I am afraid after taking into account these issues, I may not be able to increase my score.

---

> > > > ### Author Response · Authors · 2023-11-22
> > > >
> > > > Dear reviewer,
> > > >
> > > > Sorry, we did not catch the keys it in our first reply. We recall your opinion about the loss function and rewrite eq10 and its paragraph as follows:
> > > >
> > > > >Our method can be extended to most mutual information-based loss functions $\ell_(z_i, z_j)$, and we derive the enhanced loss function $L_m$ integrated multiview strategy as follows :
> > > > $$L\_m(\mathbf{z}\_1, \mathbf{z}\_2, \ldots, \mathbf{z}\_n) \triangleq \sum\_{i=1}^n I(\mathbf{z}\_{\text{mean}}, \mathbf{z}\_i) \geq \sum\_{i=1}^n \ell(\mathbf{z}\_{\text{mean}}, \mathbf{z}\_i)$$
> > > > where  $\(\mathbf{z}\_{\text{mean}} = \frac{1}{n}\sum\_{i=1}^n \mathbf{z}\_i\)$. For instance, if we consider the loss function of SimCLR, $l(z_i,z_j)$, it is often regarded as lower bounds of the mutual information (MI) between two variables(Chen et al.,
> > > > 2020; Oord et al., 2018), which establishes a connection with and benefits from our strategy.
> > > >
> > > > For your second concerns about comparison,
> > > >
> > > > As previously stated, direct comparison between SwAV (architecture + multi-cropping strategy) and our paper (multi-views strategy) is challenging. Therefore, in Table 1, we compared different strategies within the same architecture, and in the paragraph below Table 2, we contrasted our strategy applied to previous architectures  **before and after** versus SwAV (architecture + multi cropping), showing significant improvements (matches or outperforms). We acknowledge the necessity of broader comparisons—across each architecture with varying strategies
> > > > While it's reasonable to infer from existing data and theories that our method outperforms multi cropping in most baselines, we hope you understand it is currently challenging within our limited timeframe . Nevertheless, we believe our current comparisons already offer a promising direction for the field.
> > > >
> > > > Ting Chen, Simon Kornblith, Mohammad Norouzi, and Geoffrey Hinton. A simple framework for
> > > > contrastive learning of visual representations. In International conference on machine learning,
> > > > pp. 1597–1607. PMLR, 2020
> > > > Aaron van den Oord, Yazhe Li, and Oriol Vinyals. Representation learning with contrastive predic-
> > > > tive coding. arXiv preprint arXiv:1807.03748, 2018

---

### Official Review · Reviewer_z9y7 · 2023-11-01

**Soundness:** 2 fair
**Presentation:** 4 excellent
**Contribution:** 2 fair
**Rating:** 5
**Confidence:** 3

**Summary:**

The paper introduces a "plug-and-play" approach to multi-positive-views learning, seamlessly integrating with existing two-view self-supervised learning (SSL) architectures. The authors challenge traditional assumptions about multiview learning and explore its complexities. The proposed method incorporates multiple positive views to enhance traditional SSL models, improving accuracy and speed across various benchmarks and SSL architectures.

**Strengths:**

- This paper explores the complexities of multi-positive-views learning and provides an alternative way to understand multiview learning.
- Extensive experiments support the effectiveness of multiview learning.
- The paper is well-organized and easy to follow.

**Weaknesses:**

- Although the proposed strategy (Eq. 11) is an alternative way for multiple positive view contrastive learning, its novelty is limited.
- Extensive experiments are conducted. However, I can hardly find insights different from previous multiple positive view contrastive learning methods.
- In Table 2, the training epochs for each setting are not clear. If all methods share the same training epoch, the comparison is not fair since 4-view models observe more data than 2-view models.

**Questions:**

- Could you please highlight unique insights different from existing multi-positive view methods?
- In Table 2, do all the methods share the same training epochs? If yes, could you please conduct additional 2-view experiments with double training epochs for fairness?
- In Figure 7, could you please explain why GPU usage decreases as the number of views increases?

---

> ### Author Response · Authors · 2023-11-19
> **Reply to reviewer z9y7 (Part 1)**
>
> Dear Reviewer,
>
> Thank you very much for your insightful questions. In response, we have conducted additional experiments and updated our draft accordingly. We hope that these enhancements, along with our answers to your queries, will effectively address and resolve confusion. The updated part has been underlined green.
>
>
> Before delving into the main topic, we summarize major multiview strategies in recent years. Our aim is to clarify misconceptions in this area, which, despite seeming under-researched, indeed has several important gaps. The key multiview papers have been summarized as follows:
>
>
> | Name    | Year | MultiView Strategy Analysis                                                                      | Underlying Motivation                         |
> |---------|------|--------------------------------------------------------------------------------------------------|----------------------------------------------|
> | VICRegL[1] | 2022 | Multiple sub-views from 2 'main views' contrasted via distance weighting                         | Multi-view: Distance-based feature vector learning |
> | DINO[2]    | 2021 | Multi Cropping: Sum of loss functions for 2 'main views' and multiple sub-views via Multi Cropping | Averaging teacher and student models, Multi-view: Global and local feature matching |
> | SwAV[3]    | 2020 | Also Multi Cropping                                                                              | Cluster-based, Multi-view: Global and local feature matching |
> | CMC[4]     | 2019 | Any number of views, aggregate of pairwise loss functions                                       | Based on CPC, Multi-view: Providing data sufficiency value |
>
> Additionally, MIB[5] introduced a multi-view loss function based on the Information Bottleneck principle, While MIB did not extend beyond two views successfully, it was the first theoretical work to discuss multiview strategies, laying the groundwork for our research.
>  ###  Response to W1, W2, Q1
>
>  The multiview methods from these articles outlined in the table, in a strict sense, are more akin to intuitive designs for enhancing existing features, rather than being comprehensive strategies. In section 3.2, we discussed why two views (and other multiview strategies)will not work. A true multiview strategy, which goes beyond the theoretical limitation of two views, necessitates:
>
>
> 1. **Firstly, not all multiview strategies benefit from advantages of "multiview" we introduced in the paper**: We have discussed the intrinsic invisible and view-specific bias that occur in two views case. However, this also exists in most of the previous multi-view designs.
>
> 2. **Independent Discussion on Multiview Strategy**: This includes a focus on robustness and generalization capabilities. We conducted extensive experiments across various datasets and models, engaging in a thorough comparative analysis of different strategies.
> 3. **Interpretability**: As a crucial aspect of deep learning, our approach delineates a clear and logical progression from the 'why' to the 'how' of multiview strategies. Our loss function capitalizes on the multiview framework to stabilize the minimum sufficient bias, representing a significant advancement beyond a simple summation.
> 4. **Scalability**: Our method's adaptability to multiple baselines not only enhances accuracy but also helps mitigate the risk of model collapse.
>
> In conclusion, for researchers seeking to enhance or develop architectures utilizing multiview strategies, our study provides a substantial theoretical framework and compelling empirical evidence.
>
> ### Response to W3, Q2: Experiment Setup Enhancements
> We acknowledge the points raised and have accordingly augmented our experimental setup in three key respects:
>
> 1. **Equal Training Epochs**: To ensure a fair comparison of model performance, we maintained uniform training durations across all models. The outcomes are detailed in Table 2.
> 2. **Equal Data Exposure with Consistent Training Views**: To remove potential biases due to varying view exposures, we adopted consistent training views across all models. Detailed results can be found in Table 3.
> 3. **Efficiency in Achieving Specific Convergence Accuracy**: This metric provides valuable insights into the relative time efficiency of our method in comparison to others. Detailed information is available in Table 3.
>
> For more detailed explanations, please refer to the third point of our response to Reviewer K5DW.

---

> ### Author Response · Authors · 2023-11-19
> **Reply to reviewer z9y7 (Part 2)**
>
> ### Response to Q3: GPU Usage
>
> In an ideal scenario, a rudimentary formula for GPU usage decomposition is:
> $$\text{GPU Usage} = N_{\text{views}} \times (C_{\text{augmentation}} + C_{\text{forward}}) + C_{\text{loss}} + C_{\text{backward}} + C_{\text{update}}$$
> This formula indicates that the model's weight and batch size have a linear influence on GPU Usage. Under consistent configurations, GPU usage for each computational task should remain nearly constant. However, our review of debug data from previous experiments revealed some anomalies, possibly due to environmental factors, system load, or randomness.
>
> To ensure a more uniform and reliable testing environment, we repeated each test five times for every view count, discarding outliers and calculating the mean. This approach is based on the assumption that GPU usage for the same process should be consistent. We have updated our results in Figure 7.
>
> Reference:
>
> [1] Bardes et al., 2022b "Vicregl: Self-supervised learning of local visual features."
>
> [2] Caron et al., 2021 "Emerging properties in self-supervised vision transformers."
>
> [3] Caron et al., 2020 "Unsupervised learning of visual features by contrasting cluster assignments."
>
> [4] Tian et al., 2020a "Contrastive multiview coding."
>
> [5] Federici et al., 2020 "Learning robust representations via multi-view information bottleneck."

---

> > ### Comment · Reviewer_z9y7 · 2023-11-22
> > **Response by Reviewer z9y7**
> >
> > Thank you for addressing some of my concerns in your response. However, my primary issue remains regarding the novelty of the approach, particularly when compared to existing multi-view methods. The experiments in Table 1 tend to highlight the effectiveness of your approach compared with other multi-view methods. However, the experimental setup lacks clarity, which is a significant issue given the importance of this comparison. Additionally, the choice of the CIFAR-10 dataset for evaluating SSL methods is unconventional. Considering these points, I decided to maintain the original score.

---

> > > ### Author Response · Authors · 2023-11-22
> > >
> > > Dear Reviewer,
> > >
> > > Thank you for your reply. However,
> > >
> > > 1. **On Novelty**: While our method may appear straightforward, we argue that innovation lies in its advancement and importance to the field. Without this work, future research might overlook multi-view strategies, defaulting to Multi Cropping without a solid theoretical foundation, thus potentially conflating multi-view contributions with architectural ones.
> > >
> > > 2. **On Experiment Configurations**: We respectfully disagree with the comment. Detailed experimental setups for each experiment are clearly documented in the appendices, and all intermediate data are available on our anonymous GitHub. We believe our method's simplicity makes the configurations more suited for replication and reference, rather than consuming extensive space in the main text.
> > >
> > > 3. **On Dataset Selection**: We also find this comment to be unjustified. Numerous SSL studies are based on CIFAR-10, and for our focus on generalization, we've employed five datasets (CIFAR-10, CIFAR-100, STL-10, ImageNette, TinyImageNet). While using ImageNet for SOTA architectures is standard, it shouldn't limit SSL to this dataset alone.